Review Article

# Asgard archaea: have we found our microbial ancestors?

Christa Schleper [ID][1,✉] & Thiago Rodrigues-Oliveira [ID][1,2]

## Abstract

**The discovery of Asgard archaea about a decade ago has greatly reshaped our understanding of archaeal evolution and the origin of eukaryotes. Asgards are currently thought to be the closest pro-karyotic relatives of eukaryotes and to represent the archaeal host lineage that participated in the endosymbiotic event leading to the first eukaryotic cell. The presence of numerous eukaryotic signature proteins in Asgard genomes supports this view and provides important insights into the deep evolutionary roots of eukaryotic cellular complexity. However, the close relationship between archaea and eukaryotes had been observed for decades, based on features that are shared in different molecular processes. This review discusses the discovery of Asgard archaea in the broader context of archaeal molecular and cellular biology and highlights how earlier findings foreshadowed their emergence. Primarily targeted at newcomers to the field, the review provides an overview of evolutionary innovations across the Archaea domain and discusses molecular and cellular features of cultivated Asgard strains in light of previous archaeal research.**

**Keywords** Archaea; Eukaryogenesis; Evolution; Eukaryotes; Metagenomics
**Subject Categories** Evolution & Ecology; Microbiology, Virology & Host Pathogen Interaction

## Introduction

The discovery of Asgard archaea, which started only around 10 years ago, has tremendously influenced our view of archaeal evolution and that of eukaryotes. Asgards are currently considered the closest known prokaryotic relatives of eukaryotes. They are thought to have contributed to the host of the endosymbiosis that occurred with a bacterium (the pre-mitochondrion) to form the first eukaryote about two billion years ago. Hundreds of genes in Asgard archaea, previously only found in eukaryotes, seem to confirm this hypothesis and shed light on the deep origins of eukaryotic features and molecular machineries. The discovery of Asgard archaea also raises novel hypotheses about the driving forces and mechanisms that gave rise to the emergence of the first eukaryotic cell.

It is a pity that Carl Woese, the discoverer of the Archaea, could not witness the Asgard detection and missed them by just a few years. Woese sensed quickly after his finding of Archaea being a separate, third domain, that these organisms would give clues about eukaryogenesis, and he wrote several articles about early evolutionary scenarios (e.g. Woese, 1987, 2004).

Although archaeal genomes are small and circular like those of bacteria, it has already become clear since the 1980s that all archaea share astonishing similarities with eukaryotes in their molecular machineries involved in transcription, replication, translation, DNA repair, and in other complexes, indicating a shared evolutionary ancestry. This has attracted many molecular biologists over the past 40 years to study archaeal information processing. In light of the findings of these first four decades of intense molecular biological research on archaea, it becomes understandable why the recent discovery of Asgard archaea has had such a tremendous impact on the research field and beyond. As was stated in many papers and reviews, the archaea research community was almost anticipating to find a lineage that was even closer to eukaryogenesis and has transferred the archaeal inventions to eukaryotes.

This review is intended to place the discovery of Asgard archaea into the bigger context of archaeal biology and to show to what extent the close evolutionary relationship with eukaryotes was recognized well before their discovery.

The review is meant primarily for newcomers to the field of archaeal molecular and cellular biology who wish to get an overview of the evolutionary innovations found throughout the whole Archaea domain. We are often asked if there is a review that could serve as a kick start into the archaea field. But there is no single one, because the domain of archaea is huge and diverse, as are its metabolisms and cell biology. This article is meant as a first orientation to the field of Archaea with an emphasis on molecular and cellular features that were known before the year 2015, but which foreshadowed the discovery of Asgards, a lineage closer to eukaryotes than all other archaea ever investigated before. The review also discusses molecular and cellular features of the currently cultivated Asgard strains in light of these earlier discoveries.

We refer the readers to other review articles for further exciting aspects of archaea, for example on metabolism and ecological distribution (Offre et al, 2013; Baker et al, 2020; Adam et al, 2017), their adaptations to extreme environments (Valentine, 2007), their

[1]Department of Functional and Evolutionary Ecology, University of Vienna, Djerassiplatz 1, Vienna 1030, Austria. [2]Present address: Department of Biology, Aarhus University, Section of Microbiology, Ny Munkegade 114, Aarhus 8000, Denmark. ✉E-mail: christa.schleper@univie.ac.at

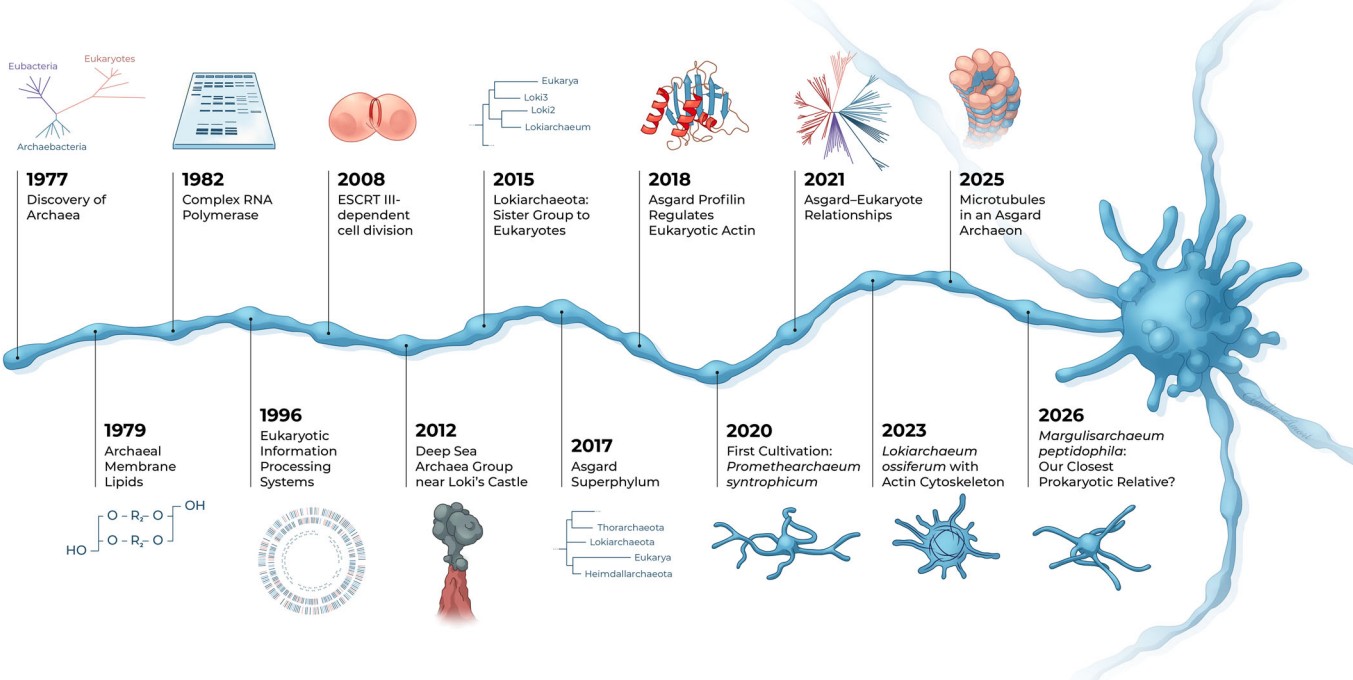

**Figure 1. Timeline of some major discoveries in the archaeal field.**

A few highlights from the first recognition of Archaea to the cultivation of the first Asgard representatives and their cellular structures are shown. Symbols indicate the respective methods used.

viruses (Prangishvili et al, 2017) and defense systems (Zink et al, 2020; Makarova et al, 2020), cell biology (van Wolferen et al, 2022), the comparative genomics of Asgard archaea and their eukaryotic signature proteins (Zaremba-Niedzwiedzka et al, 2017; Liu et al, 2021) as well as in-depth-discussions on different evolutionary and metabolic models for eukaryogenesis (Donoghue et al, 2023; Lopez-Garcia and Moreira, 2020), just to name a few.

Archaeal classification and taxonomy are currently subject to frequent revisions and parallel taxonomies spread (Guy and Ettema, 2011; Rinke et al, 2021; Hug et al, 2016). To facilitate orientation within the existing literature, we use the most commonly adopted terminology throughout this review, while also indicating the new GTDB nomenclature for higher ranks where needed for clarity.

## From discovery to diversity

Nearly 50 years ago, Carl Woese analyzed the ribosomal RNA of prokaryotes when he unexpectedly recognized that some of the investigated organisms were as different from all other bacteria as they were from eukaryotes (Woese and Fox, 1977) (Fig. 1. He first dubbed them archaebacteria, but in 1990 introduced the concept of the three domains *Archaea*, *Bacteria* and *Eukarya* (Woese et al, 1990), because it became increasingly clear that this deep phylogenetic split was supported by many cellular features. Since the late 1970s, the unique cell membranes of *Archaea* consisting of isoprenoid-derived side chains that are linked by an ether bond to a glycerol-1-phosphate were uncovered, whereas bacteria and eukaryotes have fatty acids ester-linked to a glycerol-3-phosphate (Woese et al, 1978; Tornabene and Langworthy, 1979). In addition, archaeal lipids can form a monolayer by linking the opposing hydrophobic side chains together. The division of all living organisms into those that contain ether-linked lipids with branched side chains (archaea) and those that have ester linkages and fatty acids (bacteria and eukaryotes) has often been referred to as the lipid divide (Koga et al, 1998). This divide has puzzled researchers trying to integrate it into their various evolutionary scenarios to explain the rise of eukaryotes, but it has become less sharp recently as some archaea have the genetic capacity to form ester lipids (including some Lokiarchaea) and some bacteria ether links (Villanueva et al, 2017). It will be very interesting to reveal the nature of lipids from diverse Asgard archaea, once more biomass and cleaner cultures can be obtained.

In parallel, the group of Otto Kandler recognized early that the cell walls of archaea are quite diverse and not like those of bacteria (König and Kandler, 1979). Only a few archaeal methanogenic organisms carry a peptidoglycan cell wall, that is, however, chemically distinct from bacterial murein. Most other archaea carry a proteinaceous, almost crystalline S-layer, while some have a pseudomurein cell wall and an S-layer sheet on top (Albers and Meyer, 2011). The early finding of Otto Kandler made him a vivid proponent of Carl Woese's suggestion that archaea are indeed members of a separate, third domain (Woese et al, 1990). Interestingly, like a few other archaea (e.g., *Thermoplasma acidophilum* (Darland et al, 1970), some of the so far cultivated Asgard archaea do not seem to have a regular cell wall, but rather a more irregular coating of so far unknown composition (see below).

Among the groups of organisms that Woese originally recognized as *Archaea* were methanogens (methane producers) as

well as organisms from hot environments or hot and acidic places, i.e. (hyper)thermophilic and thermoacidophilic archaea, as well as halophiles, i.e. salt-adapted organisms (see also symbols in Fig. 2. Their different adaptations and metabolisms were subjected to many studies and despite the special growth conditions, excellent genetic models were developed for representative species of each of these three groups, including *Haloferax volcanii, Halobacterium salinarum/NERC, Methanococcus maripaludis, Methanosarcina acetivorans, Thermococcus kodakarensis, Pyrococcus furiosus, Sulfolobus acidocaldarius,* and *Saccharolobus ssp* (Leigh et al, 2011). As of today, more than 600 archaea, many (but not all) from extreme environments, have been obtained as pure cultures. Several of them define some of the physical limits of life on Earth. The record holders for the hottest temperature where cells were reported to divide, *Methanopyrus kandleri* (122 °C, (Takai et al, 2008)) and the lowest pH optimum (pH 0.7, *Picrophilus oshimae* (Schleper et al, 1995) are for example defined by archaea. Upon the discovery of Archaea as a separate group, the phylogenetic tree was quickly populated by novel lineages, as pioneers like Karl Stetter and Wolfram Zillig went out to extreme places to search for more thermophilic lineages of this novel domain (Stetter, 2013). Novel strains were also cultured from high-salt environments, while methanogenic archaea were uncovered from virtually any anoxic habitat. Biological methane formation on Earth from $CO_2$ or small organic acids is solely performed by methanogenic archaea, who span the widest temperature ranges from arctic temperatures to the hottest environments, but are most widespread in anoxic wetlands and sediments as well as in ruminants (Liu and Whitman, 2008) and also occur in the human gut of about half the population (Guillaume et al, 2020). Only at the beginning of this century, another ubiquitously distributed archaeal group, autotrophic ammonia-oxidizing archaea, were cultivated, which are found in virtually any aerobic habitat and play also an important role in the global nitrogen cycle and the carbon cycle (Konneke et al, 2005; Leininger et al, 2006; Brochier-Armanet et al, 2008). They are the only aerobic archaea that are found in common-place environments, including oceans, freshwater, and soils, and even on the human skin (Probst et al, 2013; Schleper and Nicol, 2010).

The phylogenetic analysis of ribosomal RNA sequences initiated by Carl Woese did not only allow the discovery of the domain *Archaea*, but it also laid the foundation for the field of microbial ecology to study the diversity of microorganisms in complex microbiomes without the need for cultivation. Until today, 16S rRNA gene sequencing is often taken as a starting point for taxonomic profiling before metagenomic techniques are used to sequence and assemble complete or almost complete genomes from environmental samples (Tyson et al, 2004). Metagenomics has led to the discovery of many new lineages and even whole new phyla within both the bacterial and archaeal domains. For example, one of the most abundant archaeal groups, the ammonia-oxidizing archaea, found in virtually all aerobic environments on Earth, was discovered by molecular techniques more than ten years before the first organisms were cultivated (DeLong, 1992; Fuhrman et al, 1992). Metagenomics also led to the discovery of the then phylum Bathyarchaeota (now TACK or Thermoproteota) (Meng et al, 2014), a widespread lineage mostly from anoxic environments of moderate temperature. The diversity of several other widespread archaeal lineages, of which we miss cultivated members until today, including e.g. many lineages of DPANN (nano-sized archaea) and Nitroso-sphaerales was discovered by metagenomics. Finally, in 2015

metagenomics led to the discovery of Lokiarchaea in deep marine sediments, initiating the Asgard research field (Fig. 1, Spang et al, 2015).

# A complex RNA polymerase transcribes a prokaryotic genome

Already in the late 1970s Wolfram Zillig and colleagues published the first purified active RNA polymerases from halophilic and thermo-acidophilic archaea, and recognized their complexity and insensitivity to some antibiotics (Zillig et al, 1979) (Fig. 2). With the isolation of more archaea, the general picture emerged that all have RNA polymerases that are as complex as those of eukaryotes with e.g. 10 out of 13 subunits being homologous (Zillig et al, 1988). Thus, archaea must have evolved this enzyme (similar to the one we carry today in our cells) quite early in evolution. These findings made Wolfram Zillig (besides Otto Kandler) the second passionate supporter of Carl Woese's proposition of Archaea being a fundamental lineage of life distinct from Bacteria and eukaryotes (Albers et al, 2013). Subsequently, several of the core transcription factors homologous to those found in eukaryotes today were described, such as TBP (TATA box binding protein), TFB (a homolog of TFIIB in eukaryotes), and often TFE (homolog of the TFIIE alpha subunit in eukaryotes), defining a minimal machinery needed for archaeal transcription initiation (reviewed in Finn Werner and Grohmann, 2011). Different from the eukaryotic counterpart, this pre-initiation complex does not require additional factors or ATP hydrolysis to form the open complex (Blombach et al, 2016). Also, no sigma factor is needed for initiation in these microorganisms in contrast to bacteria. Coherent with these findings, the core promoter elements on the DNA level are also different from those in bacteria and carry a TATA box element similar to RNA polymerase II promoters in eukaryotes (Reiter et al, 1990; Blombach et al, 2019). Transcription elongation is mediated by Spt4/5, both conserved in eukaryotes, where Spt5 is also a homolog of the bacterial factor NusG. In addition, TFS (a homolog of TFIIS in eukaryotes) helps processivity during transcription (Sanders et al, 2019). Transcription termination factor FttA (or CPSF1) is the more recently discovered element and universally present in archaea; it is an ortholog of the eukaryotic CPSF73 (Phung et al, 2013) and helps termination for genes lacking intrinsic termination sequences (Wenck and Santangelo, 2020). This demonstrates that not only the eukaryotic RNA polymerase, but also other factors associated with transcription are offsprings of early archaeal inventions (or share a common ancestry with those from archaea).

For the discussion of gene regulation in archaea, one needs to consider the prokaryotic structure of archaeal genomes. Like bacteria, archaea have small, densely packed singular and circular chromosomes with genes arranged in clusters and also often co-transcribed in operons (Allers and Mevarech, 2005). Archaeal regulatory transcription factors (aRTFs) directly bind to an operator sequence near promoter elements where they either hinder or strengthen interactions between basal TFs and promoter elements. When repressing, TFs inhibit access of the RNAP and basal TFs to the core promoter, reminiscent of bacterial systems. However, the recruitment of the transcription complex when activated might mimic eukaryotic-like mechanisms, although in eukaryotes several activator binding sites are not only located in close proximity but also at larger distances to the transcription start site (Wenck and Santangelo, 2020).

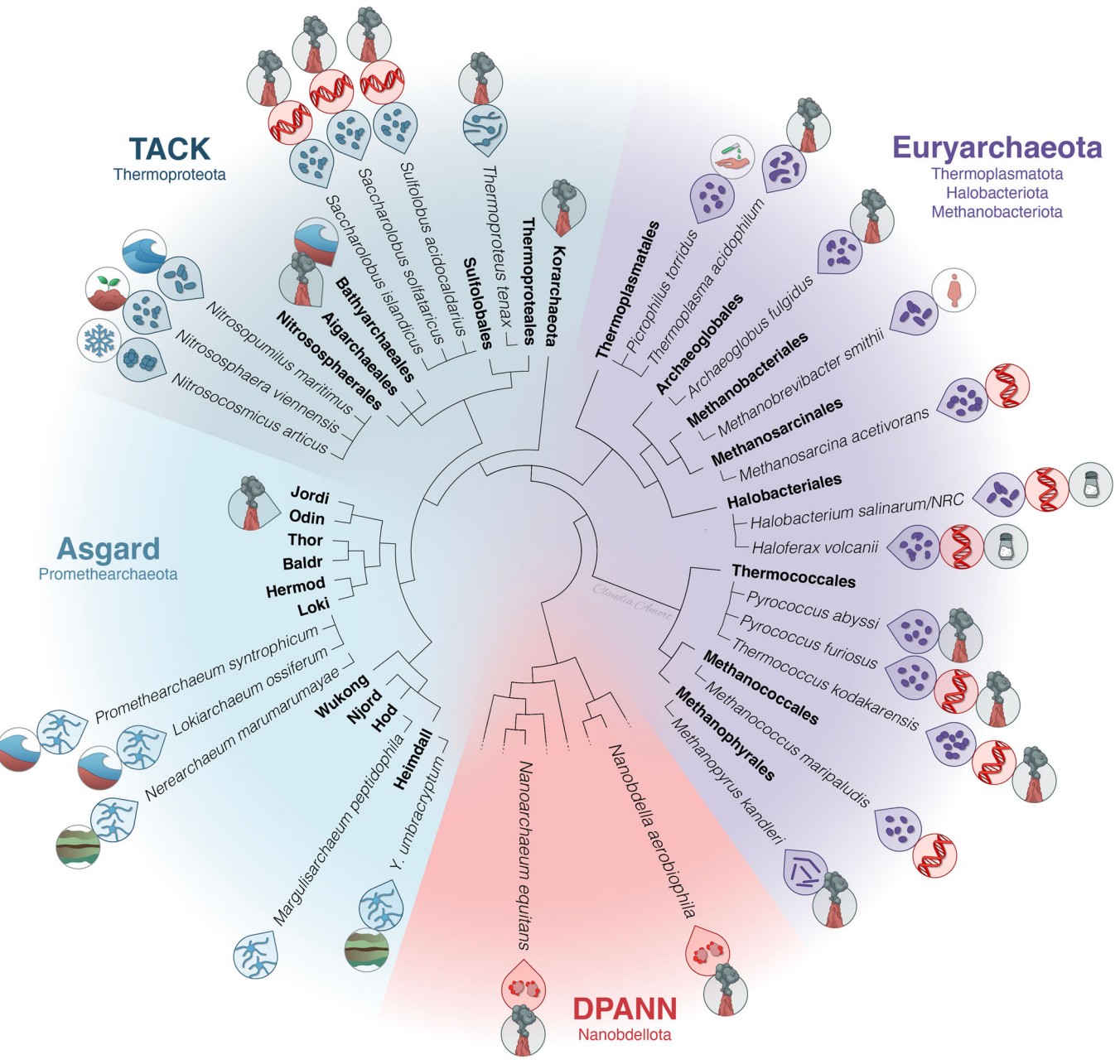

**Figure 2. Phylogenetic tree of Archaea highlighting the main archaeal groups and the species referred to in the text. Only a selection of archaea and their corresponding order levels (in bold) is represented.**

Archaeal species are all from laboratory cultures (not metagenomes). Outer symbols represent habitats (hot springs, acidic springs, sediments, salterns, ocean, soil, human), DNA symbol indidates genetic model organisms, inner symbols identify cell shapes. For higher level taxonomic ranks, both the most commonly used taxonomic nomenclature, like TACK, DPANN etc. (Hug et al, 2016; Zaremba-Niedzwiedzka et al, 2017), and the current GTDB nomenclature (in smaller font below) are presented. TACK Thaumarchaeota, Aigarchaeota, Crenarchaeota, Korarchaeota, DPANN Diapherotrites, Parvarchaeota, Aenigmarchaeota, Nanohaloarchaeota, Nanoarchaeota. In the case of Asgard archaea, we have avoided using rank endings (from phylum to class level) in the figure and refer to these lineages as Loki/Thor/Odin-archaea.

## Diversity of archaeal chromatin

Archaea have a variety of nucleoid-associated proteins (NAPs) that organize their genomes. These likely regulate DNA compaction and thus influence gene expression. Many archaea contain structural maintenance of chromosomes (SMC) proteins that seem to act as condensins in DNA compaction and organization. Takemata et al were the first to show physical and functional compartmentalization of archaeal chromosomes, of which DNA fractions, mostly bound by coalescin, a novel archaeal-specific SMC protein, are less transcriptionally active (Takemata et al, 2019). Many archaea also contain histones that are related to those in eukaryotes (Henneman et al, 2018). The proteins are highly expressed and can cover the whole genome to form superstructures (Mattiroli et al, 2017), and their blocking effects

on transcription are overcome by transcription factors TFS and Spt4/5 (Sanders et al, 2019). Stevens et al (2020) showed that some archaeal histone paralogs might act like eukaryotic histone variants, indicating that a combinatorially complex histone-based chromatin might have evolved already in archaea.

However, in methanogens, halophiles, and ammonia-oxidizing archaea (Nitrososphaerales), histones do not have N-terminal or C-terminal extensions as known from eukaryotes, and they show little posttranslational modification (Grau-Bove et al, 2022). In contrast, Asgard archaea contain an enrichment of up to nine histone genes, some of which show extended N-terminal tails with basic amino acids (mostly lysins) that might be modifiable for regulation as known in eukaryotes. Interestingly, a recent study has indicated that certain Asgard histones assemble into "closed" and "open" hypernucleosomes, revealing a conserved archaeal chromatin conformation alongside an Asgard-specific, eukaryote-like state and thereby establishing the first structure-based model of Asgard chromatin (Ranawat et al, 2025). The full role of Asgard histones in gene regulation and chromatin organization still awaits being revealed.

## The deep roots of the eukaryotic replication machinery

Shortly after the discovery of the Archaea, Forterre et al realized that some halophilic organisms are inhibited by aphidicolin, the eukaryotic inhibitor of replication (Forterre et al, 1984). The following years were then characterized by efforts to isolate archaeal DNA polymerases from hyperthermophiles to be used in the novel technology of polymerase chain reaction, PCR (Rossi et al, 1986). B family archaeal polymerases have, different from their thermostable bacterial counterparts, a 3'–5' proofreading activity and are thus less error prone and have since then been used as proofreading polymerases in PCR technology. Only later it was discovered that archaea contain another polymerase (Pol D (Ishino et al, 1998) that -at least in some species- is the only replicating protein (Cubonova et al, 2013). With the first archaeal complete genome sequences being produced in the 1990s (i.e., *Methanocaldococcus jannaschii*, back then *Methanococcus* (Bult et al, 1996), *Archaeoglobus fulgidus* (Klenk et al, 1997), *Pyrococcus horikoshii* (Kawarabayasi et al, 1998)), it became clear that the accessory replication proteins are mostly (though not exclusively) homologous to those already well-known from eukaryotes and also exhibited similar biochemical properties. Among these were the proliferating cell nuclear antigen (PCNA), replication factor C (RFC), and proteins involved in lagging-strand processing, such as DNA ligase and Fen1 (Ishino and Ishino, 2012). Similarly, the initiation of replication involves proteins that already evolved in archaea and are found in eukaryotes, in particular the minichromosome maintenance (MCM) helicase (Sakakibara et al, 2009) and the initiation protein Cdc6, also referred to as Cdc6/Orc1 (Costa et al, 2013). The archaeal helicase is a hexamer, but differently from eukaryotic cells is made up of a single protein (Barry and Bell, 2006). In *Sulfolobus*, Orc1-1 binds as a monomer to elements of the replication origin oriC1, where the ATP-bound form of the protein carries out both DNA binding and helicase recruitment functions (Samson et al, 2016). Despite differences among lineages, the archaeal replication machinery can generally be seen as a simplified, and probably ancestral form of that in eukaryotes, very

similar to the transcription machinery (Barry and Bell, 2006). A recent comparative investigation of Asgard genomes revealed a structural diversification of replisomes in different lineages, which contributed to the more sophisticated machinery in eukaryotes, likely due to horizontal gene transfers on the evolutionary path from the first towards the last Eukaryotic Common Ancestor (LECA) (Feng et al, 2025). Those transferred elements included a DNA polymerase δ-like complex in baldrarchaea, a primase complex in sif/wukong/heimdallarchaea and a RFC clamp-loader complex in lokiarchaea but also RfcS and Fen1.

Considering their small circular genomes (~0.5–6 Mbp) (Kellner et al, 2018), archaea appear as an evolutionary hybrid between bacteria and eukaryotes, with a prokaryotic-type cell and a eukaryotic-type replisome. This is also true for the number of their replication origins, which range between one (as in bacteria) to several (Samson et al, 2011), as does the ploidy of genomes (between one copy to over 20) (Ludt and Soppa, 2019; Breuert et al, 2006). Interestingly, growth and replication in halophilic (and a few other) archaea are possible when all origins of replication are deleted, and growth is even accelerated under these conditions (Hawkins et al, 2013).

While their replication machinery bears a striking similarity to that in eukaryotes, archaea show an interesting and complex mosaic of distribution of DNA repair enzymes, and also encode unique ones. For example, while a canonical bacterial mismatch repair pathway based on MutL-MutS is only found in a few lineages of mesophiles, including Asgards (White and Allers, 2018), a novel enzyme, EndoMS (for endonuclease mismatch specific), was discovered in several lineages of thermophiles to serve that role (Ishino et al, 2016). Similarly, mostly mesophilic archaea (including Asgards) seem to have adopted the (bacterial) UvrABC-based nucleotide excision repair system, while most archaea, including thermophiles (both from the Euryarchaeota and TACK) as well as mesophiles including Asgards also carry the eukaryote-type NER proteins XPF, XPD, and XPB-Bax1. Similarly, a patchy distribution of enzymes is found for the base excision and double-strand break repair systems in archaea. End-joining pathways have only been identified in specific organisms, with NHEJ being found exclusively in *Methanocella paludicola* (Bartlett et al, 2013) and MMEJ being identified *in Haloferax volcanii* (Delmas et al, 2009), (Stachler et al, 2017), and *Sulfolobus islandicus* (Zhang and Whitaker, 2018). Homologous recombination (HR) is the best-studied double-strand break repair pathway in archaea (White, 2011). While the HR presynapsis proteins Mre11 and Rad50 are conserved in archaea (Liu et al, 2016; Sung et al, 2014), the archaeal HR synapsis SSBs, although present in different groups, are structurally different. In organisms classically referred to as euryarchaea, they resemble the eukaryotic RPA, whereas in the then called crenarchaea (now within TACK or Thermoproteota, Fig. 1), they are more similar to bacterial SSBs (Lin et al, 2008; White and Allers, 2018). Notably, the HR post-synapsis helicase Hel308 is found in archaea and metazoans but not in bacteria or yeast (Woodman and Bolt, 2009; Kelman and White, 2005; White and Allers, 2018).

## Ribosomes and translation in archaea

Similar to the trends observed in other aspects of their molecular biology, the ribosomal architecture and translation machinery of

archaea also share a noteworthy number of features with those of eukaryotes. By the 1990s, it was already established that, although archaeal ribosomal proteins were organized in "bacteria-like" operons, they were more closely related at the sequence level to those of eukaryotes (Olsen and Woese, 1997). Over the following decades, additional "eukaryote-like" traits were identified in archaeal ribosomes (reviewed in (Londei and Ferreira-Cerca, 2021)). One such trait involves the structure of ribosomal RNA. In contrast to canonical prokaryotic rRNAs, most eukaryotic rRNAs contain extension segments of variable sizes inserted into the universal rRNA core (Gerbi, 1996; Bowman et al, 2020). While recent studies have detected these elements in both bacteria and archaea (Penev et al, 2020; Tirumalai et al, 2020; Stepanov and Fox, 2021), it is worth noting that large segments comparable in size to those found in eukaryotes have so far been reported only in members of the Asgard archaea (Penev et al, 2020). These extensions were used to identify Asgard cells in a growing mixed culture analyzed via Cryo-electron tomography (Rodrigues-Oliveira et al, 2023). Archaeal ribosomes not only contain the universally conserved proteins found in all domains of life (Melnikov et al, 2012), but also share a significant subset exclusively with eukaryotes while lacking any proteins uniquely shared with bacteria (Marquez et al, 2011; Ban et al, 2014).

The archaeal translation apparatus has long been established to exhibit a combination of bacterial and eukaryotic/archaeal features (Bell and Jackson, 1998). As in Bacteria, transcription and translation in Archaea occur simultaneously, and archaeal species possess polycistronic mRNAs, enabling ribosomes to perform repeated initiation cycles on the same mRNA (Benelli et al, 2017). Both Shine-Dalgarno sequences and leaderless mRNAs are found in archaea, with the former more common in euryarchaeal species and the latter predominant in the so-called crenarchaeal organisms (Ma et al, 2002; Benelli et al, 2016). Notably, Shine-Dalgarno sequences are rare in the Lokiarchaea, reflecting a noticeable difference to bacterial and other archaeal translation initiation (Imachi et al, 2020; Schmitt et al, 2020). Furthermore, genomic analyses reveal that eukaryotic initiation factors display high similarity to their homologous counterparts in archaea, including aIF1, aIF1A, aIF2, and aIF5B (Kyrpides and Woese, 1998; Benelli and Londei, 2011; Schmitt et al, 2019). Another noteworthy parallel with eukaryotes is observed in the mRNA exit tunnel: in *Pyrococcus abyssi* (a euryarchaeon), the tunnel resembles the eukaryotic type but lacks certain features, instead relying on Shine-Dalgarno-assisted AUG recognition and missing the ribosomal protein eS26, which in eukaryotes stabilizes the mRNA (Coureux et al, 2020). Ribosomal protein S26 is, however, uniquely present in eukaryotes, members of TACK archaea, and also in Asgard archaea (Schmitt et al, 2020), lending further support to the hypothesis that eukaryotic ribosomes originated from this archaeal lineage.

## Two different modes of cell division in archaea

Similar to other features previously discussed, the cell division apparatus in archaea displays noticeable diversity, with clear differences between Eury- and TACK archaea (van Wolferen et al, 2022). Most euryarchaea possess two homologs of the bacterial cell-division protein FtsZ (Ithurbide et al, 2022). The role of FtsZ1 and FtsZ2 in archaeal cytokinesis has been demonstrated in the model haloarchaeon *Haloferax volcanii* through GFP-based localization studies, and the deletion of either FtsZ homolog via genetic manipulation results in cytokinesis defects (Liao et al, 2021). Furthermore, FtsZ-based division mechanisms have also been investigated in *Methanobrevibacter smithii* (Pende et al, 2021). In both these organisms, it was revealed that a SepF homolog plays a role in anchoring the Z-ring to the membrane, similar to bacteria (Nussbaum et al, 2021; Pende et al, 2021).

In contrast to Euryarchaeota, another cell division mechanism, the Cdv system, is found in some members of TACK archaea and in Asgard archaea (Lindas et al, 2008; Zaremba-Niedzwiedzka et al, 2017) and is homologous to the eukaryotic ESCRT-III complex (reviewed in (Blanch Jover and Dekker, 2023)). The Cdv system was first described in *Sulfolobus acidocaldarius* (Lindas et al, 2008; Samson et al, 2008), where its core components, CdvA, CdvB, and CdvC, are encoded in a gene cluster essential for its viability (Zhang et al, 2018). CdvA is a protein unique to archaea that interacts with CdvB, a eukaryotic ESCRT-III homolog (Samson et al, 2011), whose paralogues have recently been implicated in membrane deformation and scission, mirroring the role of ESCRT complexes in eukaryotes (Tarrason Risa et al, 2020). CdvC, a homolog of eukaryotic Vps4 (Obita et al, 2007), appears to supply the energy required to depolymerize ESCRT-III filaments at the membrane, remodel their structure, and mediate membrane constriction and fission, also in analogy to the eukaryotic machinery (Schoneberg et al, 2018). Members of the *Nitrososphaeria* class (formerly Thaumarchaeota), also part of the TACK group, encode both the Cdv system and tubulin homologs related to FtsZ in their genomes (Bernander and Ettema, 2010; Yutin and Koonin, 2012). However, it has been suggested that the thaumarchaeon *Nitrosopumilus maritimus* relies on the Cdv system for cell division (Pelve et al, 2011). Its FtsZ-like proteins are distantly related to canonical FtsZ and lack the catalytically active domain required for this process, leaving their biological role unresolved (Aylett and Duggin, 2017). Asgard archaea also harbor genes for the Cdv machinery, where their CdvB proteins are even more similar to eukaryotic ESCRT-III than those of TACK archaea (Blanch Jover and Dekker, 2023). Interestingly, they also possess FtsZ homologs (Imachi et al, 2020; Ithurbide et al, 2022; Rodrigues-Oliveira et al, 2023), resulting in open questions on how each system contributes to the cell division process.

## The archaeal cell cycles, some with a regulated progression

Compared with bacteria and eukaryotes, the archaeal cell cycle remains poorly characterized, with the available evidence suggesting that archaea combine features of both eukaryotes and bacteria (Cezanne et al, 2024). Early studies of *Sulfolobales* revealed an ordered cell cycle resembling that of eukaryotes, characterized by a short G1 phase and a prolonged G2 phase in which cells contain two interlinked genome copies (Bernander and Poplawski, 1997; Lundgren et al, 2008). This extended G2 phase has been proposed to facilitate DNA repair (Rhind and Russell, 1998). However, other TACK archaea, such as *Nitrosopumilus maritimus*, exhibit G1 and G2 phases of similar duration (Pelve et al, 2011). Despite evidence for regulated progression, these archaea that exhibited an ordered

cell cycle lack clear homologs of core eukaryotic cell cycle regulators, including cyclins, CDKs, and the APC/C (Cezanne et al, 2024). In contrast, archaea from the Euryarchaeota branch (halophiles and methanogens) show less defined cell cycle phases and are frequently polyploid, with genome copy numbers varying by species and growth conditions (Breuert et al, 2006; Hildenbrand et al, 2011). In haloarchaea, genome replication occurs throughout the cell cycle, suggesting the absence of a distinct S phase (Zerulla and Soppa, 2014), while methanogens often display loose cell cycle control and variable ploidy (Malandrin et al, 1999; Majernik et al, 2005). No data are yet available on the cell cycle organization of Asgard archaea, reflecting their recent discovery and limited cultivation. Understanding cell cycle regulation in this group remains an important direction for future research.

## Discovery of the Asgard archaea by metagenomics

One of the groups discovered through 16S rRNA environmental surveys was the Deep Sea Archaeal Group (DSAG, formerly Deep Sea Hydrothermal Vent Crenarchaeotic Group (Takai and Horikoshi, 1999), (Takai et al, 2001), but also referred to as MBG-B (marine benthic group B (Vetriani et al, 1999)), commonly found mostly in marine sediments over the world. Especially high relative abundances of DSAG signatures appeared in certain sediment layers near the Loki's Castle hydrothermal vent system located in the Atlantic mid-ocean rift valley, representing even more than 50% of the microbial community (Jorgensen et al, 2012). The site was not directly influenced by geothermal heat, but rather by fallout from the vent system causing a distinct and sharp stratification of the sediments. Samples from a specific layer at about 2.500 mt water depth in about 2 mt deep sediment were the source for the initial discovery of the "Lokiarchaeota" (DSAG). Assembled metagenomes were sequenced for the first time, placing them as a direct sister group to eukaryotes in phylogenomic analyses (Spang et al, 2015). Equally striking was the discovery that their genomes encoded a large number of proteins previously considered unique to eukaryotes, known as Eukaryotic Signature Proteins (ESPs). These include proteins involved in membrane remodeling and vesicular trafficking, such as ESCRT system and small Ras superfamily GTPases, as well as a ubiquitinylation system and proteins linked to dynamic cytoskeleton formation, including actin, profilin, and gelsolin as well as many more functions (Spang et al, 2015). Interestingly, a few of these protein genes were detected earlier in genomes scattered throughout the archaeal tree and referred to as the "dispersed" eukaryome" (Koonin and Yutin, 2014), like e.g. the ubiquitin system in the called Aigarchaeota (Nunoura et al, 2011), as well as ESCRT-III proteins (Bernander and Ettema, 2010) and actin-related proteins (Izore et al, 2016)) in lineages of the TACK group.

Since the first discovery from the North Atlantic, additional environmental surveys have uncovered more archaeal genomes related to Lokiarchaeota. Among the newly identified groups were the Thorarchaea (Seitz et al, 2016), Odinarchaea (Zaremba-Niedzwiedzka et al, 2017), Helarchaea (Seitz et al, 2019), Heimdallarchaea (Zaremba-Niedzwiedzka et al, 2017), Gerdarchaea (Cai et al, 2020), Kariarchaea (Liu et al, 2021), Wukongarchaea (Liu et al, 2021), Hodarchaea (Liu et al, 2021; Eme et al, 2023), and

others. Collectively, these organisms are now commonly referred to as the Asgard archaea, after the home of the Nordic gods (Zaremba-Niedzwiedzka et al, 2017), with their known diversity and phylogeny continuing to expand. As taxonomy developed, Lokiarchaeota was reclassified as *Lokiarchaeia* to reflect its rank at the class level (Rinke et al, 2021; Sun et al, 2021), and Promethearchaeota has been proposed for the whole phylum, after the first cultured organism *Promethearchaeum syntrophicum* ((Imachi et al, 2020), see below). The defining feature uniting all Asgard lineages is the high abundance of shared ESPs and their monophyly, with eukaryotes in most cases (but not always) emerging close to Heimdallarchaea or Hodarchaea (Liu et al, 2021; Eme et al, 2023; Da Cunha et al, 2022).

Although first discovered in deep marine sediments, Asgard archaea are now recognized as globally distributed across a wide range of ecosystems. They occur in various anoxic sediment environments (Bulzu et al, 2019; Zou et al, 2020); Hager et al, 2025; Manoharan et al, 2019) as well as in soils and rhizospheres (Cai et al, 2021), hot springs (Zaremba-Niedzwiedzka et al, 2017), hydrothermal vents (Wu et al, 2022; Rambo et al, 2022), permafrost (Liang et al, 2021), surface oceans (Rodriguez et al, 2020), and epipelagic sediments (Appler et al, 2026). Lokiarchaea, Thorarchaea are the most widely distributed groups (Manoharan et al, 2019; Cai et al, 2021; Hager et al, 2025), whereas others appear more habitat-specific, with Odinarchaea and Njordarchaea being primarily found in high-temperature environments (Zaremba-Niedzwiedzka et al, 2017; Xie et al, 2022).

This broad distribution is mirrored by considerable metabolic diversity within this group. Genomic studies have suggested not only heterotrophic potential in the Lokiarchaea (Spang et al, 2019; Zhang et al, 2021) but also varying metabolic capabilities, including lignin and humic acid degradation, $CO_2$ assimilation, heterotrophic lactate degradation, and aromatic compound degradation (Yin et al, 2021). Helarchaea encode methyl-CoM reductase-like enzymes, suggesting potential for hydrocarbon oxidation (Seitz et al, 2019). Thorarchaea appear to share some metabolic traits with Lokiarchaea, supporting a peptide-based heterotrophy (Seitz et al, 2016; Liu et al, 2018). They also encode ribulose bisphosphate carboxylase–like proteins (without RuBisCO activity) and nearly a complete Calvin–Benson–Bassham cycle, indicating potential use of both organic and inorganic carbon (Liu et al, 2018). Wukongarchaea appear to consist of obligate hydrogenotrophic acetogens with a chemolithotrophic lifestyle (Liu et al, 2021). Heimdallarchaea possess a versatile metabolic repertoire, with evidence for a heterotrophic lifestyle via fermentation, anaerobic respiration, or even aerobic respiration (Spang et al, 2019; Appler et al, 2026). Their genomes encode a complete tricarboxylic acid (TCA) cycle supported by an electron transport chain containing V/A-type ATPase, succinate dehydrogenase, NADH–quinone oxidoreductase, and cytochrome c oxidase (Bulzu et al, 2019).

While most of the initial insights into Asgard biology stemmed from genomic data, more recent studies have begun to investigate properties of Asgard ESP proteins through biochemical experiments. Profilins from Lokiarchaea and Odinarchaea were demonstrated to adopt the typical profilin fold and interact with rabbit actin, as well as being capable to induce polymerization of mammalian actin (Akil and Robinson, 2018). Profilins from Heimdallarchaea were also shown to inhibit actin polymerization, highlighting a potential regulatory role (Survery et al, 2021), as did Thorarchaeal profilins (Inturi et al, 2022) and gelsolins

(Akil et al, 2020). ESCRT-III and VPS4 were shown to possess chromatin-binding properties (Nachmias et al, 2023), and ESCRT-III proteins from Lokiarchaea self-assembled into helical filaments, hallmarks of the ESCRT system, that bound and deformed eukaryotic-like membrane vesicles (Melnikov et al, 2025; Souza et al, 2025). Furthermore, it has also been demonstrated that Asgard ESCRT-IIIB and ESCRT-IIIA form functionally partitioned polymers whose sequential assembly and structural transitions recapitulate the conserved membrane-deformation pathway that later evolved in eukaryotes (Souza et al, 2025). Together, these findings were extensions from genome-based predictions to experimental evidence providing the first functional insights into Asgards prior to their isolation.

## Asgard archaea in laboratory culture

In 2020, Imachi and Nobu, together with colleagues from the Jamstec Institute in Japan, published the first culture of an Asgard archaeon, which they named *Candidatus* Promethearchaeum syntrophicum (Imachi et al, 2020). It is an anaerobic heterotrophic member of the Lokiarchaea which was maintained in co-culture with hydrogen-consuming microorganisms and others; by now it has been obtained as a clean co-culture with only one archaeal methanogenic partner (Imachi et al, 2024). Isolated from deep marine sediments in Japan's Kumano region, *P. syntrophicum* required a long and labor-intensive cultivation effort, mostly because of its slow growth, a common feature of deep-sea microbes, which includes long lag phases and a doubling time of 14–25 days. The morphology of *P. syntrophicum* was most striking and never observed before in archaea or bacteria: a small spherical central cell body with long—sometimes branching—protrusions (Fig. 3A–C). While many researchers had expected some internal complexity in Asgard cells based on their genomic content, the complexity of these cells seemed rather to be manifested in their cell shape.

With the cultivation of a second strain, *Candidatus* Lokiarchaeum ossiferum, which was also enriched with hydrogen-consuming microorganisms but from a shallow urban estuarine canal in Piran, Slovenia (Rodrigues-Oliveira et al, 2023) the first internal structures came to light. Although still slow-growing, *Ca.* L. ossiferum displayed a faster growth rate than *P. syntrophicum*, with a doubling time of 7–14 days and achieving up to 100-fold higher cell densities, which allowed for detailed structural investigations. Like *P. syntrophicum*, *Ca.* L. ossiferum possesses a spherical cell body with protrusions; however, these protrusions appeared more irregular, frequently branched, and often constricted, displaying bulbous structures along their length or at the tips (Fig. 3D–F). Cells were imaged with Cryo-electron tomography, identifying *Ca.* L. ossiferum cells based on characteristic expansion segments of their ribosomes. Cells were surrounded by a single membrane and complex surface structures, but no S-layer (Fig. 3F). A long-range cytoskeleton was found in the cell bodies, protrusions, and constrictions with twisted double-stranded filaments consistent with F-actin. It was confirmed to consist of the most highly expressed actin, more precisely lokiactin, highly conserved in all Asgard genomes. In a follow-up study, additional cytoskeletal elements, namely microtubules, were described (Wollweber et al, 2025). Heterologous expressions and cryo-electron tomography structures demonstrated that archaeal tubA/B form eukaryote-like heterodimers, which assembled into 5-protofilament bona fide microtubules in vitro. Non-canonical microtubules with 7-protofilaments formed through heterodimers with

an additional paralogue (Wollweber et al, 2025). Tubulins are so far only found in few Asgard genomes, thus not representing a general feature of the group. However, the study suggested a pre-eukaryotic origin of microtubules as for the actin cytoskeleton (Wollweber et al, 2025).

It is noteworthy that the first cultivated Lokiarchaea belong to the same clade, while the most environmentally widespread lineages still remain uncultured (Yin et al, 2021; Hager et al, 2025). Nevertheless, a few novel isolates have recently been obtained in culture, such as *Ca*. Margulisarchaeum peptidophila, the first hodarchaeal member, which is considered one of the closest lineages to eukaryotes (Imachi et al, 2025). In addition, a novel member of the Loki-branch (Nobs et al, 2025) and one of the Heimdall-branch (MacLeod et al, 2025) have been reported (see Fig. 2). While it may be tempting to assume from these cultures that all Asgard archaea share the same morphology, targeted fluorescence in situ hybridization experiments with environmental samples have suggested a variety of cell shapes, indicating that the morphological diversity is perhaps greater than what current cultures might suggest, or that cells adopt different shapes according to their growth state or environment (Fig. 3G–I) (Avci et al, 2022; Avci et al, 2025).

The first cultivated organisms demonstrate that fundamental features such as cell morphology, the organization of cytoskeletal elements, and many other cellular processes like cytokinesis cannot be inferred from sequence data only. Some of these features now seen in cultivated isolates were potentially crucial in eukaryogenesis. In particular, the cytoskeleton could have played an essential role in the merging of cells by phagocytosis (or another mechanism). With intracellular compartments, cells also became bigger, and a regulated intracellular transport that is also based on a cytoskeleton must have increasingly replaced mere diffusion. Equally, cell division of larger, complex cells requires a dynamic cytoskeleton.

## Models for eukaryogenesis

The origin of Eukaryotes has long been debated, and many evolutionary models have been developed reflecting ongoing scientific progress. While complex multicellularity is believed to have emerged independently several times, eukaryogenesis itself is thought to have occurred only once in evolutionary history, as eukaryotes form a monophyletic group that shares a single last eukaryotic common ancestor (LECA) (Williamson et al, 2025). However, eukaryogenesis might date back well before LECA, and might have happened more often when the first events occurred forming FECA (the first eukaryotic common ancestor) that led to a suit of follow-up events.

The idea that mitochondria and chloroplasts originated from endosymbiosis dates back to the early 20th century (Mereschkowsky, 1910; Wallin, 1883, 1922; Ward, 1883; Martin and Kowallik, 1999), was revived and popularized by Lynn Margulis in the 1960s (Sagan, 1967), and then formally proved by small subunit RNA analyses by Woese (Woese and Fox, 1977). When Woese proposed the concept of the three domains of life (Woese et al, 1990) he imagined an extra lineage that led to a proto-eukaryote (a relatively complex cell with a nucleus but no mitochondria yet) which formed a sister lineage to archaea and would later engulf a bacterial endosymbiont. Opposed to that,

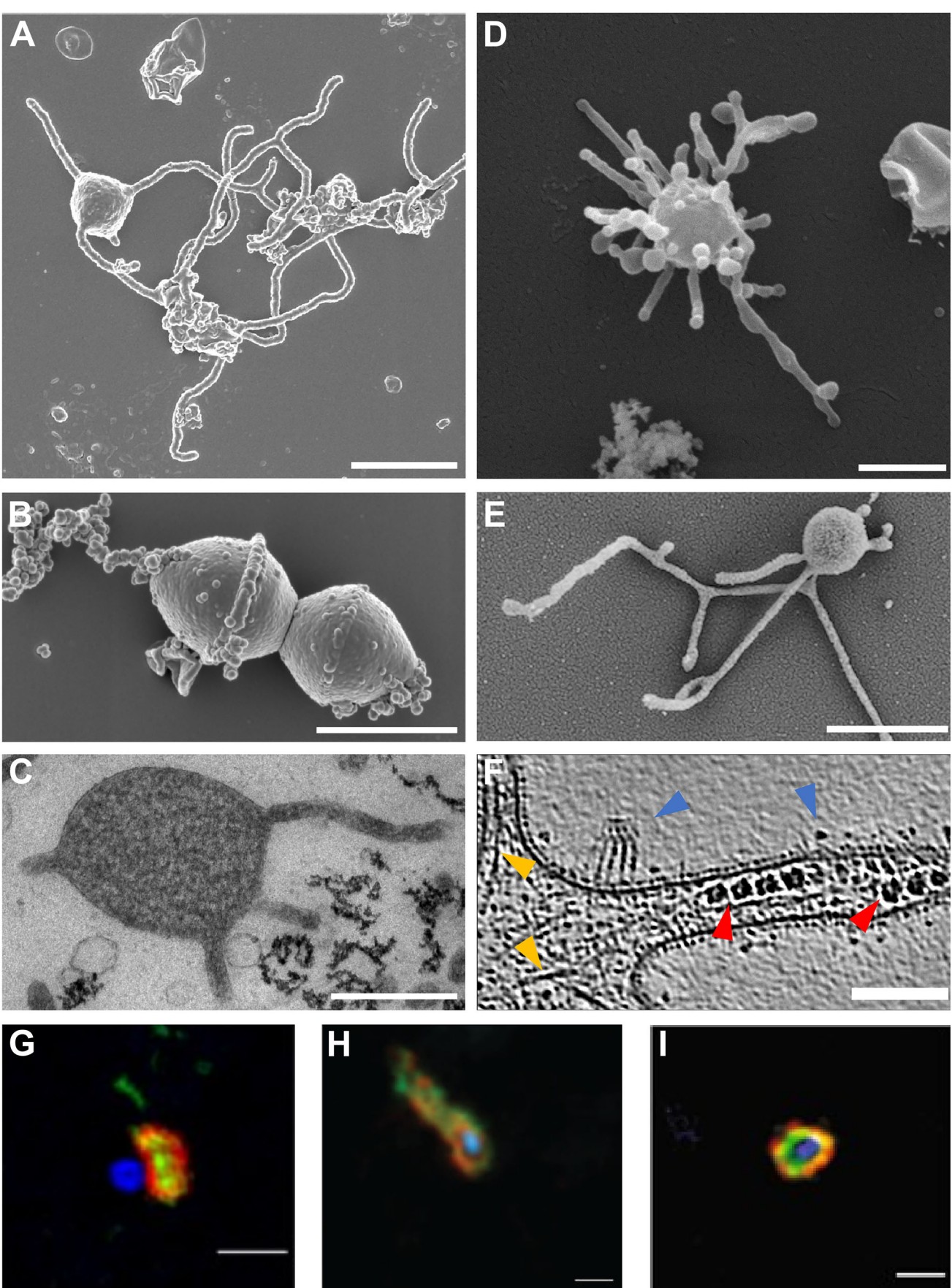

**Figure 3. Morphological features of Asgard archaea cells.**

(**A**, **B**) Scanning electron micrographs of *Promethearchaeum syntrophicum* (Imachi et al, 2020) with potentially dividing cells in (**B**). (**C**) Transmission electron micrograph of an ultrathin section of *P. syntrophicum* (Imachi et al, 2020). (**D**, **E**) Scanning electron micrographs of *Ca.* Lokiarchaeum ossiferum (personal collection). (**F**) Cryo-electron tomogram of *Ca.* L. ossiferum. Blue arrows indicate the unusual cell surface structures of this organism, orange arrows highlight actin filaments, and red arrows indicate ribosomes (adapted from (Rodrigues-Oliveira et al, 2023). (**G–I**) CARD-FISH analyses from cells collected in Aarhus Bay, Denmark of (**G**) Lokiarchaeal cells, Red—probe Lok1183, Green— Probe Lok1378, Blue—DAPI (Avci et al, 2022), (**H**) Heimdallarchaeal cells, Red—probe Heim529, Green—Probe Heim329, Blue—DAPI (Avci et al, 2022), and (**I**) Hodarchaeal cells, Red—probe Hod193, Green—Probe Hod286, Blue—DAPI (Avci et al, 2025). Scale bars: 1 µm (**A,B,D,E,G,H,I**), 500 nm (**C**), and 100 nm (**F**).

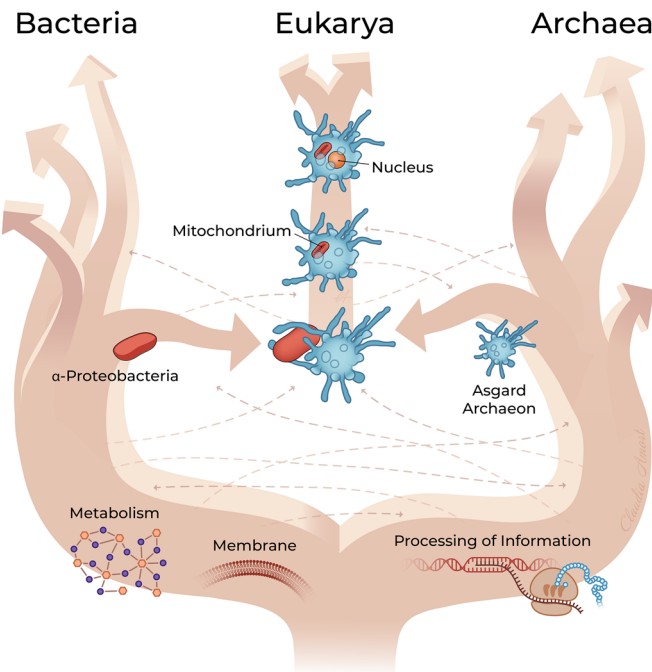

**Figure 4. Visualization of a current model about the origin of eukaryotes.**

The central role of Asgard archaea in this process is highlighted and the early split of Bacteria and Archaea as outlined in this review. Thin lines represent putative horizontal gene transfers between domains. The reader is encouraged to consult recent reviews and papers to appreciate the full breadth of ideas and hypotheses on the process of eukaryogenesis.

driven by the transfer of $H_2$ from the symbiont to the host (Martin and Muller, 1998). Another hypothesis, the Ox-Tox model, suggests instead that the symbiont consumed oxygen toxic to the host (Kurland and Andersson, 2000). These models continue to be revised as new data emerge, and in some cases also involve two bacteria (a deltaproteobacterium and an alphaproteobacterium) and an archaeal partner that produces hydrogen (Syntrophy hypothesis) (Moreira and Lopez-Garcia, 1998; Lopez-Garcia and Moreira, 2020). Currently, it is widely believed that the identity of the endosymbiont that led to mitochondria stems from alphaproteobacteria or a closely related lineage (Yang et al, 1985; Carvalho et al, 2015; Garcia et al, 2023). Although sometimes debated (Zhang et al, 2025; Da Cunha et al, 2022), phylogenomics analyses and the conservation of ESPs in Asgard archaea now overwhelmingly support these as the archaeal partner. The exact placement of eukaryotes within the Asgard lineage is still a moving target, given the regularly changing view of their diversity and phylogeny (Zhang et al, 2025; Liu et al, 2021; Eme et al, 2023). The exact placement of Eukaryotes may also not be necessarily relevant for understanding eukaryogenesis: Whereas a whole pool of evolutionary inventions which happened within the Asgard group might have contributed to the emergence of the LECA about 2 billion years ago, they might not all be found in today's living closest sister group to eukaryotes (see for example (Wu et al, 2022)). Novel evolutionary models take into account that the complexity of the cells with their protrusions might have mediated the close interaction and eventually engulfment of the symbiotic partner (Baum and Baum, 2014; Tobiasson et al, 2026; Imachi et al, 2020). The observed flexibility of the long protrusions of Asgard cells (Imachi et al, 2020; Rodrigues-Oliveira et al, 2023), but also recently observed cellular dynamics and ameboid-like cell migration, seems to support such models (Radler et al, 2025).

## The future ahead

The discovery of Asgard archaea involved a suite of observations and persistent research that started with the recognition of novel deep-branching archaea by 16S rRNA signatures. But why was their discovery as the potential eukaryotic ancestor missed for so long? In most environments preferred by Asgard archaea, they occupy no more than a few % of the microbiome diversity. So it seems it was luck that such a naturally high enrichment of more than 50% Asgard (then termed DSAG) sequences was discovered in certain layers of an unusually stratified sediment near the hydrothermal vent system Loki's castle (Jorgensen et al, 2012), as this in turn allowed the successful assembly of the very first metagenomes (Spang et al, 2015). Then the discovery of these Loki signatures in their enrichment reactors encouraged researchers to focus on the

different models were already proposed in the early 1980s when ribosomal features and RNA polymerases revealed the close archaea–eukaryote link. They suggested archaea played a direct and central role in eukaryogenesis (Henderson et al, 1984; Lake et al, 1984; Zillig et al, 1985). When phylogenomic analyses proved that amitochondriate eukaryotes derive from mitochondriate ones and are thus not from an ancestral (proto-eukaryotic) lineage that could have supplied the host part in endosymbiosis, the model in which modern eukaryotes evolved from the merger of two prokaryotic cells became again more popular. The idea that eukaryogenesis coincided with endosymbiosis of an archaeon engulfing a bacterium that later evolved into mitochondria is now the most popular (Fig. 4) (Embley and Martin, 2006; Cox et al, 2008; Guy and Ettema, 2011; Williams et al, 2012; Koonin and Yutin, 2014). Several models have attempted to explain the nature of this symbiotic interaction and the driving forces for endosymbiosis (Lopez-Garcia and Moreira, 2023). One example is the Hydrogen hypothesis, which proposes that the initial symbiosis was

cultivation of this particular lineage. In total, it took only ten years from the first identification of Lokiarchaeal genomes to the assembly of hundreds of Asgard genomes in metagenomic studies and the first cultivated organisms as well as in vitro studies of ESP proteins.

This research field has become vibrant with many activities now in different fields of evolution, including phylogenomics, comparative (meta)genomics, cultivation of syntrophic consortia and in the investigation of deep origins of eukaryotic features within Asgards. The near future will certainly reveal many more details about this still enigmatic group that is worth studying not only in the light of eukaryotic evolution but also in itself. It is already becoming possible to grow cultures to high enough biomass for biochemical investigations, and we will hopefully be able to develop genetic systems for these organisms to allow studying gene functions in vivo. Through the cultivation of more species, we will better understand which -sometimes unexpected- features of Asgards are common to the whole phylum and are thus likely to be transferred to the first eukaryotic cell and which ones are unique to certain lineages.

Could eukaryogenesis still happen today? Nobody has currently the answer to that, but Charles Darwin answered the question whether life could have originated more than once: "It is often said that all the conditions for the first production of a living organism are now present, which could ever have been present. But if (and oh what a big if) we could conceive in some warm little pond with all sorts of ammonia and phosphoric salts,—light, heat, electricity, etc. present, that a protein compound was chemically formed, ready to undergo still more complex changes, at the present day such matter would be instantly devoured, or absorbed, which would not have been the case before living creatures were formed" (Darwin, 1871).

Considering how complex and delicate the first close interaction of bacteria with an Asgard archaeon and their further evolution into LECA must have been, this view might also apply to eukaryogenesis, should it occur nowadays. Then, how likely can it be that a similar evolution would occur on other planets? These deep evolutionary questions are not new, and reach out to other disciplines, such as astrobiology and philosophy. What has changed now is the fact that we are a step closer to completing the picture, and able to tell a more comprehensive evolutionary story. With Asgard archaea as the plausible second partner in the symbiosis that led to eukaryotes, we have a narrative that can be communicated to the public better than ever before. We can tell how and why this event was so unique and fascinating, and of course, crucial for our very own existence. And not to forget, as Lynn Margulis liked to point out: "it was an event of cooperation and not competition" (Margulis and Sagan, 1986).

## Peer review information

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

## Acknowledgements

We would like to thank all colleagues who have contributed to the archaeal field over the years, for their discussions and interactions. We apologize for not having cited everyone. We thank Frederic Berger, Nevena Maslac, Philipp Radler as well as the four reviewers for their input and important comments on the manuscript. We are also indebted to Nathalia Jandl for help with formatting references. Thanks to Kai Finster for pointing out the Darwin citation. This work was supported by the Austrian Science Fund (FWF) with W1257 (Wittgenstein) and EFP 25 (Emerging Field Project) to CS.

## Author contributions

**Christa Schleper**: Conceptualization; Funding acquisition; Writing—original draft; Writing—review and editing. **Thiago Rodrigues-Oliveira**: Writing—original draft; Writing—review and editing.

## Disclosure and competing interests statement

The authors declare no competing interests.

