## [Peer Review File · The EMBO Journal]

Asgard archaea: Have we found our microbial ancestors?

Christa Schleper and Thiago Rodrigues-Oliveira

Corresponding author: Christa Schleper (christa.schleper@univie.ac.at)

Review Timeline:

Submission Date:	26th Nov 25
Editorial Decision:	17th Dec 25
Revision Received:	28th Jan 26
Accepted:	3rd Feb 26

Editor: Yehu Moran

Transaction Report:

Dear Dr. Schleper,

Thank you for submitting your manuscript for consideration by the EMBO Journal. It has now been seen by four referees whose comments are enclosed. As you will see, all of them express interest in your manuscript and are broadly in favour of publication, pending satisfactory minor revision.

Please treat these comments as suggestions. Yet, all four referees are experts in the field and I believe their comments are mostly helpful, so the more you can incorporate and take on board in a reasonable time, the better. I would especially encourage you to positively consider the comment by Referee #3 regarding adding a figure that can help newcomers to the field who are less familiar with some of the archaeal lineages you mention. I believe this can be quite helpful.

Lastly, I also include comments from our editorial assistance team that are crucial to resolve before we can officially accept your paper.

We are looking forward to receiving your revised manuscript at your earliest convenience.

Yours sincerely,

Yehu Moran
Academic Editor
The EMBO Journal

Editorial assistance team reports

MANUSCRIPT FORMAT: Please provide a .docx file with no figures, no track changes embedded in it.

ORCID ID: Please provide.

AFFILIATIONS (research institution or university vs. biotech company): employment in a biotech company should be stated in DCIS.

Keywords (up to 5): missing, please provide.

COI/DCIS: should be included with the title "DISCLOSURE AND COMPETING INTERESTS STATEMENT"

AUTHOR CONTRIBUTIONS: ok

FIGURES: Figures 1 and 3 should be turned around, Fig. 2 should not be uploaded as PowerPoint (.pptx) format, but in .eps, .tif, or .jpg format of high resolution. Fig. 2 should be provided without legend, which should only be listed in manuscript below the References.

FIGURE LEGENDS: should be placed below the References

FIGURE CALLOUTS: all callouts should be listed sequentially; missing callout for Fig. 1, please correct.

REFERENCE FORMAT: should have 10 authors listed + et al.

Additional Notes:

- Section order should be corrected: Title page - Abstract - Keywords - Introduction - Acknowledgements (if any) - Disclosure and Competing Interests Statement - References - Figure Legends - Table(s) (if any)

Referee reports

Referee #1:

This review represents an accessible, enjoyable and informative overview of progress in the archaeal field in general and Asgard in particular and is written by leading experts in the field. I think it will serve well as an introduction to this fascinating group of organisms. I particularly enjoyed the historical context in which the work was framed.

I have only a few specific comments.

Lines 155 - 157. The authors should explain that "coalescin" is a novel, archaeal specific SMC protein. Further, ClsN is enriched in transcriptionally less-active regions of the chromosome, not transcriptionally active regions as the authors state.

Lines 197 - 199. There is no evidence that archaeal Orc1/Cdc6 form ring shaped hexamers. Rather the work of Samson et al (Mol. Cell, 2016) revealed, that in *Sulfolobus*, the Orc1-1 protein binds as a monomer to elements in the origin oriC1 and this monomeric Orc1-1, when bound to ATP, functions in both DNA binding and helicase recruitment roles.

Line 281-291

CdvA, B and C are not in an operon. CdvA is a separate transcription unit from the bicitronic cdvBC genes (see Samson, Mol Cell, 2011). The authors should re-phrase this section from "Encoded in an operon" to "encoded in a gene cluster". In addition, the authors should mention that CdvC is a homolog of eukaryotic Vps4 (which they discuss later in the context of Asgard ESCRT machineries). It's a shame that the CdvC nomenclature has been propagated in the literature, it just adds confusion to an already complex literature - indeed, the archaeal Vps4 was first named as Vps4 by Obita and colleagues in 2007 (Obita et al., Nature, 449, 735-739,) a year before Bernander and colleagues chose to rename it as CdvC.

A minor point but the image in Figure 3 is awfully reminiscent of Jim Lake's Ring of Life hypothesis (proposed long before the discovery of Archaea". It would be nice to cite Lake's seminal proposal (Rivera and Lake Nature 2004 Sep 9;431(7005):152-5. doi: 10.1038/nature02848.)

I, Steve Bell, am happy to sign this report.

Referee #2:

Albeit surely not exhaustive (impossible task), this is an enjoyable and personal account that will surely be of interest to the readership of the journal.

I had quite a few comments and suggestions that I made directly on the text. The authors can retain what they feel relevant and useful. I do not need to be anonymous.

Referee #3:

This review article by Schleper and Rodrigues-Oliveira represents a timely and (mostly) comprehensive overview of the history and current state-of-the-art of research into the archaea, and Asgard archaea more particularly. This covers a lot of ground, and does so in an even-handed way. The article will be invaluable for readers intrigued by the archaea. The figures are particularly attractive and I can see them being adapted for many lectures on archaea.

Inevitably, there are omissions. Several minor, some major. I have tried to outline these below. Overall, this is a very welcome and authoritative addition to the literature on the archaea / Asgard.

Major points

1. Many different archaeal clades, phyla and species are mentioned, but there's no figure to put this in perspective. Newcomers

to the field could be very confused. What is a crenarchaeote? Euryarchaeote? TACK superphylum? How do they relate to Asgard or to each other? I feel another figure is essential.

2. Another major omission is a discussion of the archaeal cell cycle - these needs at least a paragraph to do it justice - there are many interesting parallels and differences to eukaryotes in the different archaeal lineages.

Specific points:

1. In several places, reviews are cited instead of the primary literature. The latter would be preferable, but if there's a lack of space then at least make it clear that the reference supplied is a review.
2. There is no discussion of Homologous Recombination / DSB, which should be rectified if this is a comprehensive review.
3. A general readership might be interested in the early research on archaeal CRISPR systems and the fact that archaea lack class 2 crisper (Cas9, Cas12) but are rich in type I and III.
4. Methanogens are mentioned in several places, but readers may wish to know that all methanogens are archaeal, given the environmental implications.

Minor points:

1. In general, the English could be improved a little. A few specific suggestions are given below.
2. Line 18. "It also.." - presumably this refers back to "The discovery" but the two clauses are separated by an entire paragraph, so maybe just re-state discovery in the final sentence.
3. Line 21-23. Consider referencing one or more of Woese's papers here.
4. Line 33. The phrase "evolutionary achievements" might be considered a little unscientific?
5. Line 35 "archaeal biology"
6. Line 107 "of of"
7. Line 122-3. Consider adding the original references (Bell et al.) here.
8. Line 125 "an ATP hydrolysis" - step/reaction?
9. Line 127 "these finding" -s
10. Line 130 Blombach 2019 is a review - if you want to use this perhaps make this clear.
11. Line 133 "recently discovered universally in archaea conserved" - need rephrased.
12. Line 131, 135 - settle on one spelling of -og / -ogue
13. Line 177 what does "and later also others" refer to? People or species?
14. Line 186-7 italicize species names.
15. Line 187 sp "Archaeoglobus"
16. Line 188. The phrase "which initiated biochemical studies" is confusing. Perhaps best deleted.
17. Line 213 - reference missing
18. Line 226 - references needed here, or at least a review.
19. Line 256 typo Lokiarchaeia ?
20. Line 272 Define "TACK"
21. Line 512 - best not to mention unpublished data.
22. Line 515 "asgard"
23. Line 514-524 - needs some citations, even if they are repeated.

Referee #4:

This is a timely review, that while accessible to newcomers to the field is nonetheless comprehensive and balanced. I have just a few comments.

Preamble

"It also raised novel hypotheses about the driving forces and mechanisms that gave rise to the emergence of the first eukaryotic cell"

should be

"The discovery of Asgard Archaea..."

"evolutionary achievements" - rephrase

"It is mainly addressed to newcomers " - should be "The review is meant primarily for newcomers to the field of archaeal molecular and cellular biology who..."

"inventions" - should be "innovations"

"ether bondage" -> "ether bond"

"ester-linked fatty acids to a glycerol-3-phosphate" - rephrase

"racked the brains of researchers" -> "puzzled researchers"

"It will very interesting to see what kind of lipids the Asgard archaea have..." - actually Imachi et al., showed rather convincing evidence that

'MK-D1 probably contains C20-phytane and C40-biphytanes with 0-2 rings. The MK-D1 genome encoded most of the genes necessary to synthesize ether-type lipids-although geranylgeranylglyceryl phosphate synthase was missing-and lacked genes for ester-type lipid synthesis'

"ar ρ for example from archaea" => "are archaea" or "are members of the archaea"

"i.e. thermophilic and thermoacidophilic" -> "i.e. hyperthermophilic and thermoacidophilic"

"N-terminal tales" -> |N-terminal tails"

|between one copy to >20 ref." => fix

"Lokiarchaeia" -> check that it has not since also been renamed

It is important to note that most current models of eukaryogenesis propose serial endosymbiosis, perhaps starting with deltaproteobacteria with alphaproteobacteria arriving much later, see recent work from the university of Bristol. These scenarios reject the "big bang" fusion that was popular before the cultivation of Asgard archaea. This aspect of the paper should be edited accordingly.

**Asgard archaea: Have we found our microbial ancestors?**

Christa Schleper^{1*} and Thiago Rodrigues-Oliveira^{1#}

¹University of Vienna, Dep. of Functional and Evolutionary Ecology, Djerassiplatz 1, 1030

Vienna, Austria

[#]new address: Aarhus University, Section of Microbiology, Dep. of Biology, Ny Munkegade

114, 8000, Aarhus, Denmark

*corresponding author christa.schleper@univie.ac.at

**Preamble**

The discovery of Asgard archaea which started only around 10 years ago has
 tremendously influenced our view of archaeal evolution and that of eukaryotes. Asgards,
 ~~the phylum of Promethearchaeota~~, are currently considered the closest known
 prokaryotic relatives of eukaryotes. They are thought to have contributed the host of the
 endosymbiosis that occurred with a bacterium (the pre-mitochondrion) to form the first
 eukaryote about two billion years ago. Hundreds of genes in Asgard archaea, previously
 only found in eukaryotes seem to confirm this hypothesis and shed light on the deep
 origins of eukaryotic features and molecular machineries. It also raises novel hypotheses
 about the driving forces and mechanisms that gave rise to the emergence of the first
 eukaryotic cell.

It is a pity that Carl Woese, the discoverer of the Archaea, could not witness the Asgard
 discovery and missed them just by a few years. Woese sensed quickly after his finding of
 Archaea being a separate, third domain, that these organisms would give clues about
 eukaryogenesis and he wrote several articles about early evolutionary scenarios.

Although archaeal genomes are small and circular like those of bacteria, it became
 already clear since the 1980s that all archaea share astonishing similarities with
 eukaryotes in their molecular machineries involved in transcription, replication,
 translation, DNA repair, and in other complexes. This has attracted many molecular
 biologists over the past 40 years to study archaeal information processing. In the light of
 the findings of these first four decades of intense molecular biological research on
 archaea, it becomes understandable why the recent discovery of Asgard archaea has had
 such tremendous impact on the research field and beyond. It feels as if we have always

Commented [SG1]: throughout the text: Archaea in capital letters when referring to the domain, and archaea in lower case when referring to some archaea in general

Deleted: A

Deleted: A

Commented [SG2]: I don't think this is necessary here and even confusing, you can leave it for later in the paragraph on the discovery of asgard and their new diversity/taxonomy

Commented [SG3R2]: Asgards is still the main recognized name, especially for the large audience

Deleted: i.e.

Deleted: to

Deleted: um

Deleted: so far

Deleted: d

Deleted: was not able to

Deleted: a number of

Deleted: A

Deleted: that are

Deleted: ,

Deleted: A

Deleted: also

known and waited for that one lineage to be discovered, that was more directly involved
in eukaryogenesis and transferred these evolutionary achievements to eukaryotes.

This review is intended to place the discovery of Asgard archaea into the bigger context of
archaeal biology and to show that the close evolutionary relationship with eukaryotes
was recognized well before their discovery.

We mainly address to newcomers who wish to get an overview on the evolutionary
inventions found throughout the whole Archaea domain. We are often asked if there is a
review that could serve as a kick start into the archaea field. But there is no single one,
because the domain of archaea is huge and diverse as are its metabolisms and cell
biology. This article is meant as a first orientation to the field of Archaea with an emphasis
on molecular and cellular features that were known before the year 2015, but which
foreshadowed the discovery of Asgards, a lineage closer to eukaryotes than all other
archaea ever investigated before. The review also discusses molecular and cellular
features of the currently cultivated Asgard strains in the light of these earlier discoveries.

We refer the readers to other review articles for further exciting aspects of archaea,
including metabolism and ecological distribution (Offre et al., 2013), (Baker et al., 2020),
viruses (Prangishvili et al., 2017), defense systems (Zink et al., 2020); (Makarova et al.,
2020), their adaptations to extreme environments (Valentine, 2007) the comparative
genomics of Asgard Archaea and their eukaryotic signature proteins (Zaremba-
Niedzwiedzka et al., 2017), (Liu et al., 2021) as well as in-depth-discussions on different
evolutionary and metabolic models for eukaryogenesis (Donoghue et al., 2023), (Lopez-
Garcia and Moreira, 2020).

Main text

From Discovery to Diversity

Nearly 50 years ago, Carl Woese analysed the ribosomal RNA of prokaryotes when he
unexpectedly recognized that some of the investigated organisms were as different from
all other bacteria as they were from eukaryotes (Woese and Fox, 1977). He first dubbed
them archaeobacteria, but in 1990 introduced the concept of the three domains Archaea,
Bacteria and Eukarya (Woese et al., 1990), because it became clear that this deep
phylogenetic split was supported by many cellular features. Since the 1980s it is known
that Archaea have unique cell membranes consisting of isoprenoid-derived side chains

Deleted: about

Deleted: ut

Deleted: between archaea and

Deleted: of the Asgard archaea

Deleted: It

Deleted: is

Deleted: ed

Deleted: ant

Deleted: found

Deleted: that is even

Deleted: e.g.

Commented [SG4]: here it would be nice to follow the timeline in Figure 1 (which btw is not referred to in the text)

Commented [SG5R4]: I also wonder if you have space for another figure but it would be so much better for the readership to have an updated schematic tree of archaea, to understand what you refer to in the text (for example TACK might not mean anything for the non-archaea people)

Commented [SG6R4]: a tree could also be a nice visual of all the newly discovered diversity

Commented [SG7R4]: I am sure you can redraw one from the recent literature, or we can provide one if you wish

Commented [SG8R4]: you can refer the readers here to the main three reviews on archaea diversity and taxonomy (baker nat rev micro, spang science, brochier isme) as they got out almost at the same time (it may be time for a more recent one!)

Commented [SG9]: another general suggestion is to try and cite mostly recent reviews and the major original references, as sometimes it feels a bit random

that are linked by an ether bond, to a glycerol-1-phosphate, whereas bacteria and
eukaryotes have fatty acids ester-linked to a glycerol-3-phosphate (Woese et al., 1978).
In addition, archaeal lipids can form a monolayer by linking the opposing hydrophobic
side chains together. The division of all living organisms into those that contain ether
linked lipids (archaea) and those that have ester linkages and fatty acids (bacteria and
eukaryotes) has often been referred to as the lipid divide (Koga et al., 1998). This divide
has racked the brains of researchers trying to integrate it into their various evolutionary
scenarios to explain the rise of eukaryotes. It will be very interesting to see what kind of
lipids the Asgard archaea have. Lokiarchaeal genomes suggest that they might possess
the genetic capacity to synthesize G3P-based 'chimeric lipids', which could represent a
membrane evolutionary transition stage in the archaeal-to-eukaryotic membrane shift
(Villanueva et al., 2017). However, experimental data on this topic are still missing.
In the early 1980s the group of Otto Kandler recognized that the cell walls of archaea are
quite diverse and not like bacterial peptidoglycan (ref Kandler archaeal cell walls review).
Only a few archaeal methanogenic organisms carry a peptidoglycan cell wall, but it is
chemically distinct from bacterial murein and was therefore named pseudomurein (refs
Kandler and König, one of which was recently refined (Baquero et al., 2025)), and has
also different side chains. Most other archaea carry a proteinaceous, almost crystalline
S-layer, while some have a pseudomurein cell wall and an S-layer sheet on top, or no cell
wall at all (Albers and Meyer, 2011). The early finding of Otto Kandler made him a vivid
proponent of Carl Woese's suggestion that archaea are indeed members of a separate,
third domain (Woese et al., 1990). Interestingly, like a few other archaea (e.g.
*Thermoplasma acidophilum* (Darland et al., 1970), some of the so far cultivated Asgard
archaea do not seem to have a regular cell wall, but rather a more irregular coating of so
far unknown composition (see Fig. 2f, below).
Among the groups of organisms that Woese originally recognized as Archaea were
methanogens (methane producers) as well as organisms from hot environments or hot
and acidic places, i.e. thermophilic and thermoacidophilic archaea, as well as
halophiles, i.e. salt-adapted organisms. Their different adaptations and metabolisms
were subjected to many studies and despite the special growth conditions, excellent
genetic models exist today for representative species of each of these three groups
(Dyall-Smith, 2009), (Leigh et al., 2011). As of today, more than 600 archaea, mostly from

Deleted: age

Deleted: ester-linked

Commented [SG10]: if possible, it would be nice here to say that the ether/ester divide is now less sharp as some archaea have been shown to have ester links and some bacteria ether links. What remains the fundamentals of the lipid divide is actually the G1P/G3P split, as so far all archaea have G1P and all bacteria/eukarya have G3P

Commented [SG11]: why is it not possible to do it now that there are pure cultures? Or if it's known that the cultured archaea have classical G1P, then maybe say it here (or later when talking about the cultured ones)

Deleted: 's group

Deleted: A

Commented [SG12]: here you can chose to refer to the original old references or to a recent review by sonja albers (I think there is a more recent one than that of 2011, check it out)

Deleted: murein

Deleted: consists of different building blocks than

Commented [SG13]: Baquero 2025 did not refine the structure of archaeal PG

Formatted: Strikethrough

Formatted: Strikethrough

Formatted: Strikethrough

Deleted: have both,

Deleted: ¶

Formatted: Font: Italic

Deleted: outer

Commented [SG14]: sorry if I'm picky but it is always better to refer to the figures in their order, here referring to 2f before all the others seems a bit odd. I think you don't need to do it now, you can refer to it later when actually talking about figure 2

Deleted: structures

Commented [SG15]: it would be nice to the newcomers to know the names of these models

Commented [SG16]: are these two references necessary here?

Deleted: Until t

Deleted: A

134 extreme environments, have been obtained as pure cultures. Several of them define
some of the physical limits of life. The record holders for the hottest temperature where
cells have been seen to divide (122 °C, (Takai et al., 2008)) and the lowest pH optimum
(pH 0.7, *Picrophilus oshimae*, (Schleper et al., 1995) are for example from archaea.
The phylogenetic analysis of ribosomal RNA sequences did not only allow the discovery
of the domain Archaea, but it also laid the foundation for the field of microbial ecology to
study the diversity of microorganisms in complex microbiomes without the need of
cultivation. It is often taken as a starting point before metagenomic techniques are used
to sequence and assemble complete or almost complete genomes from environmental
samples (Perez-Cobas et al., 2020). Metagenomics has led to the discovery of many new
lineages and even whole new phyla within both the bacterial and archaeal domains. For
example, one of the most abundant archaeal groups, the ammonia oxidizing archaea,
found in virtually all aerobic environments on Earth were discovered by molecular
techniques first (DeLong, 1992), (Fuhrman et al., 1992). Metagenomics also led to the
discovery of Bathyarchaeota (Meng et al., 2014), a widespread lineage mostly from anoxic
environments, and finally in 2015 to the discovery of Asgard archaea in marine sediments
(Spang et al., 2015).

A Complex RNA Polymerase Transcribes a Prokaryotic Genome

Already in the late 1970s Wolfram Zillig and colleagues published the first purified active
RNA polymerases from halophilic and thermoacidophilic archaea, and recognized their
complexity and insensitivity to some antibiotics (Zillig et al., 1979). With the isolation of
more archaea, the general picture emerged that all have RNA polymerases that are as
complex as those of eukaryotes with 10 out of 13 subunits being homologous (Gehring et
al., 2016), (Jun et al., 2011). Thus, archaea must have evolved this enzyme (similar to the
one we carry today in our cells) quite early in evolution. These findings made Wolfram
Zillig (besides Otto Kandler) the second passionate supporter of Carl Woese's
proposition of Archaea being a fundamental lineage of life distinct from Bacteria and
eukaryotes (Albers et al., 2013). Subsequently, several of the core transcription factors
homologous to those found in eukaryotes were described, such as TBP (TATA box binding
protein), TFB (a homolog of TFIIB in eukaryotes) and often TFE (homolog of the TFIIE alpha
subunit in eukaryotes), defining a minimal machinery needed for archaeal transcription

Commented [SG17]: I always heard from Forterre that this temperature could never be reproduced and that the actual limit would be more around 110°. Also, please say which species.

Commented [SG18]: here it would be the good moment to dismantle the widespread idea of archaea as just extremophiles. Mention that they are found everywhere in mesophilic environments, water column, and even in the animal and human gastrointestinal tract. This info would be surely interesting for newcomers to the field.

Formatted: Font: Italic

Deleted: e

Deleted: for phylogeny has not only led to

Deleted: that still today relies on 16S rRNA gene sequencing ...

Deleted: bacteria and archaea

Commented [SG19]: I think here it would be fair to cite the seminal paper by jill banfield: Tyson, G.W., Chapman, J., Hugenholtz, P., Allen, E., Ram, R.J., Richardson, P., Solovyev, V., Rubin, E., Rokhsar, D., and Banfield, J.F. (2004) Community structure and metabolism through reconstruction of microbial genomes from the environment. *Nature*, 428, 37 – 43.

Deleted: that are

Commented [SG20]: there are many others no? why mentioning specifically the Bathys? is the reference correct for the discovery and naming of the phylum? otherwise mention how it was called at the time?

Deleted: of

Deleted: b

Deleted: also

Deleted: A

Deleted: colleagues

Deleted: of

Deleted: a

Deleted: beside

Deleted: b

Commented [SG21]: orthologous is technically correct but homologous is easier to understand for the general audience

Deleted: orthologous

Deleted: like

[revised manuscript text omitted]

Deleted: mostly coalescin-bound

Deleted: ¶
¶

Commented [SG24]: or you want to say all Euryarchaeota?

Deleted: euryarchaeota (

Deleted: and

Deleted:)

Deleted: they

Deleted: to that

Commented [SG25]: stupid rule, but all numbers before ten should be in letters

Deleted: 9

Deleted: from

Moved down [2]: (Forterre et al., 1984)

Moved (insertion) [2]

Deleted: the

Deleted:

Formatted: Font: Italic

Deleted: *u*

Formatted: Font: Italic

Formatted: Font: Italic

it became clear that the accessory replication proteins are mostly (though not
exclusively) homologous to those well-known from eukaryotes and also exhibited similar
biochemical properties. Among these were the proliferating cell nuclear antigen (PCNA),
replication factor C (RFC), and proteins involved in lagging strand processing such as
DNA ligase and Fen1 (Ishino and Ishino, 2012). Similarly, the initiation of replication
involves proteins found in eukaryotes, in particular the minichromosome maintenance
(MCM) helicase (Sakakibara et al., 2009) and the initiation protein Cdc6, also referred to
as Cdc6/Orc1 (Costa et al., 2013). The archaeal helicase is a hexamer, but different from
eukaryotic cells is made up from a single protein (Barry and Bell, 2006). Archaeal cells
use AAA⁺ proteins related to the largest subunit of ORC, Orc1 and to Cdc6, with varying
numbers of subunits that form the ring-shaped hexamer that binds DNA (Barry and Bell,
2006). Despite differences among lineages, the archaeal replication machinery can
generally be seen as a simplified, and probably ancestral form of that in eukaryotes,
similar to the transcription machinery (Barry and Bell, 2006). A recent comparative
investigation of Asgard genomes revealed a structural diversification of replisomes in
different lineages, which contributed to the more sophisticated machinery in eukaryotes
probably by horizontal gene transfers on the evolutionary path towards LECA (Feng et al.,
2025). Those include a DNA polymerase δ -like complex in Baldrarchaeia, a primase
complex in Sif/Wukong/Heimdallarchaeia and an RFC clamp-loader complex in
Lokiarchaeales, but also RfcS and Fen1.
Considering their small circular genomes (~0.5 to 6 Mbp) (Kellner et al., 2018), archaea
appear as an evolutionary hybrid between bacteria and eukaryotes, with a prokaryotic-
type cell and a eukaryotic-type replisome. This is also true for the number of their
replication origins, which range between one (as in bacteria) to several (Samson et al.,
2011) as does the ploidy of genomes (between one copy to >20 ref.). Interestingly, growth
and replication in halophilic (and a few other) archaea is possible when all origins of
replication are deleted, and it even accelerates growth in these conditions (Hawkins et
al., 2013).
While their replication machinery bears a striking similarity to that in eukaryotes, archaea
show an interesting mosaic of distribution of DNA repair enzymes, and also encode
unique ones. For example, while a canonical bacterial mismatch repair pathway based
on MutL-MutS is only found in few lineages of mesophiles, including Asgards (White and

Deleted: proteins in

Deleted: e.g.

Deleted:

Deleted: ing

Deleted: archaeal

Deleted: an

Deleted: , that

Commented [SG26]: if first time, spell out the acronym

Commented [SG27]: I think it's a class and that's the correct name

Deleted: .

Deleted: a

Commented [SG28]: try being consistent with names, for example just mention the class level (Lokiarchaeia) unless you want to specifically refer to classes orders families etc.

Deleted: , previously considered to be transmitted vertically (Feng et al., 2025)

Deleted: ¶

Deleted: prokaryotic

Deleted: A

Deleted: thus

Deleted: to be like

Formatted: Highlight

Deleted: even in the absence of any

Deleted: ¶

Deleted: E

Deleted: A

[revised manuscript text omitted]

Deleted: Another

Deleted: ,

Deleted: , is the Cdv system, an apparatus

Formatted: Font: Italic

Commented [SG31]: attention, Thaumarchaeota never mentioned before, you can do it when introducing the AOA. Also maybe precise that they are not a phylum anymore and have been renamed Nitrosopumilales (if I'm not wrong, I'm a bit lost sorry...)

Deleted: shown

Formatted: Font: Italic

Commented [SG32]: they have two if I remember correctly

Deleted: is

Deleted: s

Deleted: its

**A Dispersed Eukaryome and Phylogenomic Studies prophesied the Asgard Archaea**
The last 40 years of molecular biological and biochemical studies have firmly established
what was hypothesized already in the 1980s, namely that Archaea and Bacteria, beside
sharing fundamental features (in metabolism, cell size, genome organisation etc.), must
have separated very early in evolution into two major lineages (Makarova et al., 1999), as
it is so well documented in the archaeal information processing machineries.

Deleted: ¶

Commented [SG33]: this paragraph is a bit odd, it is much smaller than the others and a bit repetitive, maybe remove and reduce as an introduction to the next paragraph?

Deleted: a

Deleted: b

Deleted:

Commented [SG34]: why this ref? maybe a ref is not even necessary here

Deleted: been

Commented [SG35]: what does it mean?

In addition to the similarity of eukaryotic information processes, other proteins equally
known as hallmarks of eukaryotic cells were discovered in some lineages of archaea.
They include: histone proteins in Euryarchaeota and Thaumarchaeota, the ubiquitin
system in Aigarchaeota (Nunoura et al., 2011), ESCRT-III proteins (Bernander and Ettema,
2010) and an actin-related protein (crenactin (Izore et al., 2016)) in lineages of the TACK
superphylum. This “dispersed” eukaryome (Koonin and Yutin, 2014) foreshadowed a
lineage which would combine all these features and be at the origin of eukaryotes.
Together with phylogenomic studies that increasingly showed a possible eukaryotic root
within the archaea (Williams et al., 2013), these findings have inspired the search for
novel deep branching lineages in remote environments (Gribaldo et al., 2010), (Koonin,
2010).

The Discovery of the Asgard Archaea Lineages by Metagenomics

In the decades that followed the proposal of Archaea as a separate domain of life in 1990
(Woese et al., 1990), environmental 16S rRNA gene surveys and the assembly of
environmental metagenomes greatly expanded our view of archaeal diversity, with
approximately 30 archaeal phyla now identified in the literature (Baker et al., 2020),
(Tahon et al., 2021).

One of the groups discovered through 16S rRNA environmental surveys was the Deep Sea
Archaeal Group (DSAG, formerly Deep Sea Hydrothermal Vent Crenarchaeotic Group
(Takai and Horikoshi, 1999), (Takai et al., 2001), but also referred to as MBG-B (marine
benthic group B (Vetriani et al., 1999)), commonly found mostly in marine sediments over
the world. Especially high relative abundances of DSAG signatures appeared in certain
sediment layers near the Loki’s Castle hydrothermal vent system located in the Atlantic
mid-ocean rift valley, representing more than 50% of the microbial community (Jorgensen
et al., 2012). The site was not directly influenced by geothermal heat, but rather by fallout
from the vent system causing a distinct and sharp stratification of the sediment. These
samples from sediments about 2.500 m deep in about 2 m deep sediment were the
source for the initial discovery of the 'Lokiarchaeota' (DSAG). Assembled metagenomes
were sequenced for the first time, placing them as a direct sister group to eukaryotes in
phylogenomic analyses (Spang et al., 2015). Even more striking was the discovery that
their genomes encoded a large number of proteins previously considered unique to

Deleted: found in some but not all lineages of archaea that are ...

Commented [SG36]: this has already been said earlier

Deleted: e.g.

Deleted: e

Commented [SG37]: this has already been said just above

Deleted: in one

Deleted: became the ancestral lineage

Deleted: to

Deleted: (Williams et al., 2013)

Commented [SG38]: A congruent phylogenomic signal places eukaryotes within the Archaea.
Williams TA, Foster PG, Nye TM, Cox CJ, Embley TM. Proc Biol Sci. 2012 Dec 22;279(1749):4870-9. doi: 10.1098/rspb.2012.1795. Epub 2012 Oct 24. PMID: 23097517

Commented [SG39R38]:

Proc Natl Acad Sci U S A

. 2015 May 26;112(21):6670-5.

doi: 10.1073/pnas.1420858112. Epub 2015 May 11.

The two-domain tree of life is linked to a new root for the Archaea

Kasie Raymann 1, Céline Brochier-Armanet 2, Simonetta

Gribaldo 3

Affiliations

[1]

Commented [SG40R38]: if you want to cite more th ... [2]

Commented [SG41]: I don't know about the koonin r ... [3]

Commented [SG42R41]: here I would put the review ... [4]

Deleted: 4- 1

Formatted: Strikethrough

Deleted: a

Formatted: Strikethrough

Formatted: Strikethrough

Deleted: its

Formatted: Strikethrough

Formatted: Strikethrough

Formatted: Strikethrough

Formatted: Strikethrough

Formatted: Strikethrough

Formatted: Strikethrough

Deleted: representig

Commented [SG44]: check this sentence

Deleted: r

Deleted: r

Deleted: -

Deleted: typical of and rather

eukaryotes, known as Eukaryotic Signature Proteins (ESPs). ~~These include proteins~~
involved in membrane remodeling and vesicular trafficking, such as ESCRT system and
small Ras superfamily GTPases, as well as a ubiquitinylation system and proteins linked
to dynamic cytoskeleton formation, including actin, profilin, and gelsolin and many more
(Spang et al., 2015).

Deleted: ,

Deleted: which

Deleted: those

529 Since then, additional environmental surveys have uncovered more archaeal genomes
with characteristics similar to Lokiarchaeota. Among the newly identified groups were
Thorarchaeia (Seitz et al., 2016), Odinararchaeia (Zaremba-Niedzwiedzka et al., 2017),
Helarchaeales (Seitz et al., 2019), Heimdallarchaeia (Zaremba-Niedzwiedzka et al.,
2017), Gerdarchaeales (Cai et al., 2020), Kariarchaeaceae (Liu et al., 2021),
Wukongarchaeia (Liu et al., 2021), Hodarchaeales (Liu et al., 2021), (Eme et al., 2023),
and others. Collectively, these organisms are now commonly referred to as the Asgard
archaea (Zaremba-Niedzwiedzka et al., 2017), with their known diversity and phylogeny
continuing to expand. As taxonomy developed, Lokiarchaeota was reclassified as
Lokiarchaeia to reflect its rank at the class level (Rinke et al., 2021), (Sun et al., 2021) and
Promethearchaeota is now the official taxonomic name of the whole phylum, after the

Commented [SG45]: you see why this is redundant with the previous small paragraph where you mention the same proteins

first ~~cultured~~ organism *Promethearchaeum syntrophicum* (Imachi et al., 2020), see
below). The defining feature uniting all Asgard lineages is the abundance of shared ESPs
and their monophyly, with Eukaryotes in most cases (but not always) emerging close to
the Heimdallarchaeia or the Hodarchaeales (Liu et al., 2021), (Eme et al., 2023), (Da Cunha
et al., 2022).

Commented [SG46]: why some are referred to as classes, some as orders, and some as families?

Commented [SG47]: I think it comes better here than in the beginning of the review

Deleted: described

Deleted: ir

Deleted: groups of

Deleted: from within Asgard

Deleted: ly

Deleted: linked

Although first discovered in deep marine sediments, Asgard archaea are now recognized
as globally distributed across a wide range of ecosystems. Studies have identified them
in various anoxic sediment environments (Bulzu et al., 2019), (Zou et al., 2020), (Hager et
al., 2025), (Manoharan et al., 2019) as well as in soils and rhizospheres (Cai et al., 2021),
hot springs (Zaremba-Niedzwiedzka et al., 2017), hydrothermal vents (Wu et al., 2022),
(Rambo et al., 2022), permafrost (Liang et al., 2021), surface oceans (Rodriguez et al.,
2020), and epipelagic sediments (Appler et al., 2024). Lokiarchaeia, Thorarchaeia and
also Heimdallarchaeia are the most widely distributed groups (Manoharan et al., 2019),

Commented [SG48]: isn't this paper supporting the 3D tree of life?

(Cai et al., 2021), (Hager et al., 2025), whereas others appear more habitat-specific, with
Odinararchaeia and Njordarchaeia~~ja?~~ being primarily found in high temperature
environments (Zaremba-Niedzwiedzka et al., 2017), (Xie et al., 2022).

Commented [SG49]: ?

Deleted: a

Deleted: les

This broad distribution is mirrored by considerable metabolic diversity within this group.
Genomic studies have suggested not only heterotrophic potential in the Lokiarchaeia
(Spang et al., 2019), (Zhang et al., 2021) but also varying metabolic capabilities, including
lignin and humic acid degradation, CO₂ assimilation, heterotrophic lactate degradation,
and aromatic compound degradation (Yin et al., 2021). Helarchaeia? encode methyl-
CoM reductase-like enzymes reminiscent of those in butane-oxidizing archaea,
suggesting potential for hydrocarbon oxidation (Seitz et al., 2019). Thorarchaeia appear
to share some metabolic traits with Lokiarchaeia, where their genomes are also rich in
extracellular peptidases, peptide uptake systems, membrane transporters, and
intracellular proteases, supporting a peptide-based heterotrophy (Seitz et al., 2016), (Liu
et al., 2018). They also encode ribulose biphosphate carboxylase-like proteins (without
RuBisCO activity) and nearly a complete Calvin-Benson-Bassham cycle, indicating
potential use of both organic and inorganic carbon (Liu et al., 2018). Wukongarchaeia
appear to consist of obligate hydrogenotrophic acetogens with a chemolithotrophic
lifestyle (Liu et al., 2021). Heimdallarchaeia possess a versatile metabolic repertoire,
with evidence for a heterotrophic lifestyle via fermentation, anaerobic respiration or
aerobic respiration (Spang et al., 2019). Their genomes encode a complete tricarboxylic
acid (TCA) cycle supported by an electron transport chain containing V/A-type ATPase,
succinate dehydrogenase, NADH-quinone oxidoreductase, and cytochrome c oxidase
(Bulzu et al., 2019).

While most of the initial insights into Asgard biology stemmed from genomic data, more
recent studies have begun to investigate properties of Asgard proteins through
biochemical experiments. Profilins from Lokiarchaeia and Odinararchaeia were
demonstrated to adopt the typical profilin fold and interact with rabbit actin, as well as
being capable to induce polymerization of mammalian actin (Akil and Robinson, 2018).
Profilins from Heimdallarchaeia were also shown to inhibit actin polymerization,
highlighting a potential regulatory role (Survery et al., 2021). Thorarchaeal profilins (Inturi
et al., 2022) and gelsolins (Akil et al., 2020) were similarly shown to regulate eukaryotic
actin. ESCRT-III and VPS4 were shown to possess chromatin-binding properties
(Nachmias et al., 2023), and ESCRT-III proteins from Lokiarchaeia self-assembled into
helical filaments, hallmarks of the ESCRT system, that bound and deformed eukaryotic-
like membrane vesicles (Melnikov et al., 2025), (Souza et al., 2025). Furthermore, it has

Deleted: a

Formatted: Subscript

Deleted: ales

Commented [SG50]: would a figure with a detailed tree of asgard help the reader here to navigate through all these names?

Deleted: Lokiarchaeal and Odinararchaeal p

Deleted: and

Deleted: polymerized in the presence of Asgard profilins

Deleted: Heimdallarchaeal

Deleted: p

Deleted: d

also been demonstrated that Asgard ESCRT-IIIB and ESCRT-IIIA form functionally
partitioned polymers whose sequential assembly and structural transitions recapitulate
the conserved membrane-deformation pathway later evolved in eukaryotes (Souza et al.,
2025). Together, these findings were extensions from genome-based predictions to
experimental evidence providing the first functional insights into Asgards prior to their
isolation.

Deleted: elaborated

Deleted: first

Deleted: first

Deleted: even before organisms were cultivate

Deleted: d.

Asgard Archaea in Laboratory Culture

In 2020, Imachi and Nobu together with colleagues from the Jamstec Institute in Japan
published the first culture of an Asgard archaeon, which they named Candidatus
Promethearchaeum syntrophicum (Imachi et al., 2020). It is an anaerobic heterotrophic
member of the Lokiarchaeia class which was maintained in co-culture with hydrogen-
consuming microorganisms and others; by now it has been obtained as a clean co-
culture with only one archaeal methanogenic partner (Imachi et al., 2024). Isolated from
deep marine sediments in Japan's Kumano region, P. syntrophicum required a long and
labor-intensive cultivation effort, mostly because of its slow growth, a common feature
of deep-sea microbes, which includes long lag phases and a doubling time of 14–25 days.

Deleted: Hiroyuki

Deleted: Masaru

Formatted: Font: Not Italic

Deleted: and other

Deleted: (Imachi et al., 2020)

Deleted: but

Deleted: this

Deleted: organism

Deleted: d

The morphology of P. syntrophicum is most striking and never observed before in archaea
or bacteria: a small spherical central cell body with long sometimes branching-
protrusions (Figure 2a-c). While probably most researchers had expected some internal
complexity in Asgard cells based on their genomic content, the complexity of these cells
seemed rather to be manifested in their cell shape.

Deleted: A

Commented [SG51]: you forgot a-c labels in the new version of the figure

With the cultivation of a second strain, Candidatus Lokiarchaeum ossiferum, which was
also enriched with hydrogen consuming microorganisms but from a shallow urban
estuarine canal in Piran, Slovenia (Rodrigues-Oliveira et al., 2023) the first internal
structures came to light. Although still slow-growing, Ca. L. ossiferum displayed a faster
growth rate than P. syntrophicum, with a doubling time of 7–14 days and achieving higher
cell densities, which allowed for detailed structural investigations. Like P. syntrophicum,
Ca. L. ossiferum possesses a spherical cell body with protrusions; however, these
protrusions appeared more irregular, frequently branched, and often constricted,
displaying bulbous structures along their length or at the tips (Figure 2d-f). Cells were
imaged with Cryo-electron tomography, identifying Ca. L. ossiferum cells based on

Commented [SG52]: in the case of candidatus, only Ca is in italic and the species names not, at least I think this is the rule

Formatted: Font: Italic

Formatted: Font: Italic

Formatted: Font: Italic

Formatted: Font: Italic

Deleted: it

Deleted: in Ca. L. ossiferum

Deleted: c

Deleted: '

Deleted: '

Deleted: on the basis of

659 characteristic expansion segments of their ribosomes. Cells were surrounded by a single
membrane and complex surface structures but no S-layer (Figure 2f). A long-range
cytoskeleton was found in the cell bodies, protrusions and constrictions with twisted
double-stranded filaments consistent with F-actin. It was confirmed to consist of the
most highly expressed actin, more precisely lokiactin, highly conserved in all Asgard
genomes. In a follow-up study, additional cytoskeletal elements, namely microtubules
were described (Wollweber et al., 2025). Cryo-electron microscopy structures
demonstrated that AtubA/B form eukaryote-like heterodimers, which assembled into 5-
protofilament bona-fide microtubules in vitro. Non-canonical microtubules with 7-
protofilaments formed through heterodimers with an additional paralog (Wollweber et
al., 2025). Tubulins are only found in very few Asgard genomes, thus not representing a
general feature of the group. However, the study suggested a pre-eukaryotic origin of
microtubules as for the actin cytoskeleton (Wollweber et al., 2025).

It is noteworthy that all three currently cultivated Lokiarchaea belong to the same clade,
while the most environmentally widespread lineages still remain uncultured (Yin et al.,
2021), (Hager et al., 2025). A few novel isolates have recently been obtained in culture,
such as *Margulisarchaeum peptidophila*, the first member of the Hodarchaea,
considered as one of the closest lineages to eukaryotes (Imachi et al., 2025). While it may
be tempting to assume that all Asgard archaea share this morphology, hybridization with
environmental samples have suggested a variety of cell shapes, indicating that the
morphological diversity is perhaps greater than what current cultures might suggest, or
that cells adopt different shapes according to their growth state (Fig. 2g-i, (Avci et al.,
2022; Avci et al., 2025).

Although genomic analyses provide substantial insight, offering a basis for protein
expression studies, detailed phylogenomics analyses, and metabolic predictions, the
first investigations on Asgard cells already remind us that genomes alone cannot capture
everything. Important and fundamental features such as cell morphology, the
organization of cytoskeletal elements and other cellular processes like cytokinesis
cannot be inferred from sequence data only. Likewise, growth-dependent morphological
changes, dynamic cellular behaviors like motility can only be understood through direct
observation of living cells.

Commented [SG53]: tomography?

Commented [SG54]: what are these proteins which have not been introduced before, say how do we know that the tubulins identified by cryo correspond to these proteins

Commented [SG55R54]: or maybe this is the name you gave to asgard tubulins? in any case please clarify

Deleted:

Deleted: non-canonical microtubules

Deleted: s

Deleted: it does

Commented [SG56]: family? order?

Deleted: lokiarchaeal groups

Commented [SG57]: which ones? it would be nice to mention them and give references (I'm not aware of all these cultures!)

Deleted: ¶

Deleted: like e.g.

Deleted: a

Deleted: ales

Formatted: Font: Italic

Deleted: that are

Deleted: hybridisation

Commented [SG58]: what does it mean?

Deleted: over

Deleted: ,

Deleted: division

Deleted: only

**Models for Eukaryogenesis**

The origin of Eukaryotes has long been debated, and many evolutionary models were
developed reflecting ongoing scientific progress. While complex multicellularity is
believed to have emerged independently several times, eukaryogenesis itself is thought
to have occurred only once in evolutionary history, as eukaryotes form a monophyletic
group that shares a single last eukaryotic common ancestor (LECA) (Worden et al., 2015),
(Hug et al., 2016). However, eukaryogenesis might date back well before LECA, when the
first events occurred forming FECA (the first eukaryotic common ancestor) that led to a
suit of follow up events.

The idea that mitochondria and chloroplast originated from endosymbiosis dates back to
the early 20th century (Mereschkowsky, 1910), (Wallin, 1883; 1922), (Ward, 1883),
(reviewed in Martin and Kowallik, 1999), was revived and popularized by Lynn Margulis in
the 1960s (ref!!), and then formally proved by small subunit RNA analyses by Woese (ref
1977). When Woese proposed the concept of the 3 domains of life (Woese et al., 1990)
he imagined a lineage that led to a proto-eukaryote (a relatively complex cell with a
nucleus but no mitochondria yet) which formed a sister lineage to archaea and would
later engulf a bacterial endosymbiont. Notably, this line of reasoning that archaea played
a central role in eukaryogenesis was already proposed in the early 1980s when ribosomal
structures and RNA polymerases revealed a close eukaryote-archaea link (Henderson et
al., 1984), (Lake et al., 1984), (Zillig et al., 1985). Later on, phylogenomic analyses proved
that amitochondriate eukaryotes derive from mitochondriate ones, converging to a
model in which modern eukaryotes likely emerged from the merging of two prokaryotic
cells, with a bona-fide archaeon engulfing a bacterium that later evolved into
mitochondria (Embley and Martin, 2006), (Cox et al., 2008), (Guy and Ettema, 2011),
(Williams et al., 2012), (Koonin and Yutin, 2014) (Figure 3). Several models have
attempted to explain the nature of this symbiotic interaction and the driving forces for
endosymbiosis, such as the Hydrogen hypothesis, which proposes that the initial
symbiosis was driven by transfer of H₂ from the symbiont to the host (Martin and Muller,
1998). Another hypothesis, the Ox-Tox model, suggests instead that the symbiont
consumed oxygen toxic to the host (Kurland and Andersson, 2000). These models
continue to be revised as new data emerge, and in some cases also involve two bacteria
(a deltaproteobacterium and an alphaproteobacterium) and an archaeal partner that

Deleted:

Deleted: Eukaryogenesis

Deleted: (Adl et al., 2012), (Burki et al., 2020), (Koonin, 2010). Consequently, all

Deleted: are considered to

Commented [SG61]: why these references specifically? I don't think Hug is relevant here, you can cite papers on the tree of eukaryotes (there are a few recent ones):

A robustly rooted tree of eukaryotes reveals their excavate ancestry

Kelsey Williamson, Laura Eme, Hector Baños, Charley G. P. McCarthy, Edward Susko, Ryoma Kamikawa, Russell J. S. Orr, Sergio A. Muñoz-Gómez, Bui Quang Minh, Alastair G. B. Simpson & Andrew J. Roger

Nature volume 640, pages 974–981 (2025)Cite this article

Deleted: of

Deleted: and

Commented [SG62]: I think this is important to say!

Deleted: Carl

Deleted: common ancestry

Deleted: with

Moved (insertion) [3]

Deleted: ¶
However,

Deleted: such '

Deleted: '

Deleted: have never

Deleted: been found until today and phylogenomic studies increasingly converged in

Deleted: one partner,

Moved up [3]: Notably, this line of reasoning that archaea played a central role in eukaryogenesis was already proposed in the early 1980s when ribosomal structures and RNA polymerases revealed a close eukaryote-archaea link (Henderson et al., 1984), (Lake et al., 1984), (Zillig et al., 1985).¶

Deleted: ,

Deleted: .

Deleted: ¶

Commented [SG63]: You can cite here the review by moreira and garcia presenting all the different models, it'll be useful for the readership.

Deleted: was transferred

Formatted: Subscript

Deleted: ,

Deleted: while

Deleted: that was thought to be

Deleted: a

770 produces hydrogen (Syntrophy hypothesis) ((Lopez-Garcia and Moreira, 2020),
(Donoghue et al., 2023)). Currently, it is widely believed that the identity of the
endosymbiont that led to mitochondria stems from alphaproteobacteria or a closely
related lineage (Yang et al., 1985), (Carvalho et al., 2015), (Wang and Luo, 2021), see
(Lopez-Garcia and Moreira, 2020). Although sometimes debated (Zhang et al., 2025), (Da
Cunha et al., 2022), phylogenomics analyses and the conservation of ESPs in Asgard
archaea now overwhelmingly support them as the archaeal partner. The exact placement
of eukaryotes within the Asgard lineage is still a moving target given the regularly changing
view of their diversity and phylogeny (refs). It may also be not necessarily relevant for
understanding eukaryogenesis, because whereas a whole pool of evolutionary inventions
which happened within the Asgard group might have contributed to the emergence of the
LECA about 2 billion years ago, they might not all be found in today's living closest sister
group to eukaryotes (see for example (Wu et al., 2022)). Novel evolutionary models take
into account that the complexity of the cells with their protrusions might have mediated
the close interaction and eventually engulfment of the symbiotic partner (Baum and
Baum, 2014), (Imachi et al., 2020). The observed flexibility of the long protrusions of
Asgard cells (Imachi et al., 2020), (Rodrigues-Oliveira et al., 2023), but also recently
observed cellular dynamics and amoeboid-like migration seem to support such models
(unpublished).

Open questions and the future ahead

The discovery of Asgard archaea involved a suite of observations and persistent research
that started with the recognition of novel deep branching archaea as shown by 16S rRNA
signatures in the deep ocean. Perhaps it was luck that a naturally high enrichment was
subsequently found in certain layers of an unusually stratified sediment near the
hydrothermal vent system Loki's castle, as this in turn allowed the successful assembly
of metagenomes, which revealed novel eukaryotic features in an archaeal sister clade to
eukaryotes. The discovery of these Loki signatures in their enrichment reactors
encouraged researchers to focus on the cultivation of this particular lineage. In total, it
took only ten years from the first identification of Lokiarchaeota to the assembly of
hundreds of Asgard genomes to the first cultivated organisms.

Commented [SG64]: cite also the original 1998 paper

Deleted: s

Commented [SG65]: why this ref?

Commented [SG66]:

PLoS Biol

. 2023 Nov 8;21(11):e3002374.

doi: 10.1371/journal.pbio.3002374. eCollection 2023 Nov.

Components of iron-Sulfur cluster assembly machineries are
robust phylogenetic markers to trace the origin of
mitochondria and plastids

Pierre Simon Garcia 1 2 , Frédéric Barras 2 , Simonetta

Gribaldo 1

Affiliations

PMID: 37939146 PMCID: PMC10631705 DOI:

10.1371/journal.pbio.3002374

Deleted: ive

Deleted: of that lineage

Deleted: eukaryotic signature proteins

Deleted: the latter as the origin of

Deleted: position

Deleted: controversial

Deleted: (Zhang et al., 2025) and

Deleted: but

Commented [SG67]: explain what this paper says precisely

Deleted: e.g.

Deleted: has

Commented [SG68]: now you can cite your biorxiv I
think?

Commented [SG69R68]: and also the one of Buzz Baum
on internal vesicles

Commented [SG70]: indeed there is only one open
question here?

Deleted: a

Deleted: . Because

Deleted: in turn

Deleted: 10

Deleted: recognition of a favorite Loki

Deleted: place via the

818 This research field has become vibrant with many activities now in different fields of
 819 evolution, including phylogenomics, comparative (meta)genomics, cultivation of
 820 syntrophic consortia and in the investigation of deep origins of eukaryotic features within
 821 Asgards. The near future will certainly reveal many more details about this enigmatic
 group that is worth studying not only in the light of eukaryotic evolution but also in itself.
 We will soon be able to grow cultures to high enough biomass for biochemical
 investigation, and we will hopefully be able to develop genetic systems for these
 organisms to allow studying gene functions *in vivo*. Through cultivation of more species
 we will better understand which -sometimes unexpected- features of Asgards are
 unifying for the whole phylum and which ones are unique to certain lineages.
 Could eukaryogenesis still happen today? Nobody has currently the answer to that, but
 Charles Darwin would perhaps answer like this: 'If it happens today, the organism would
 not survive for very long, because it would be outcompeted by others that evolved earlier
 and are already better adapted to the environment'. At least this is approximately what
 he replied in a letter to the question about whether life could have originated more than
 once (Darwin, 1871). Considering how complex the first interaction of a bacterium with
 an Asgard archaeon and its further evolution into LECA must have been, this
 consideration might also apply to eukaryogenesis. Then, how likely can it be then that a
 similar evolution would occur on other planets? These deep evolutionary questions are
 not new, and reach out to other disciplines, such as astrobiology and philosophy. What
 has changed now is the fact that we are somehow closer to closing the circle, and able to
 tell a more complete evolutionary story. With Asgard archaea as the plausible partner in
 the symbiosis that led to eukaryotes, we have a narrative that can be communicated to
 the public better than ever before. We can tell how and why this event was so unique and
 fascinating and of course crucial for our very existence. And not to forget, as Lynn
 Margulis liked to point out: "it was an event of cooperation and not competition" (Margulis
 and Sagan, 1986).

 **Figure legends:**

Figure 1: Timeline of some discoveries in the archaeal field, from the recognition of
 Archaea to the cultivation of the first Asgard representatives and their cellular structures.

- Deleted: prokaryotes
- Deleted: It
- Deleted: in the near future
- Deleted: by itself, not only -but also-
- Deleted: s
- Deleted: which
- Deleted: s to
- Deleted: learn
- Deleted: ,
- Deleted: ¶
- Commented [SG71]: I would remove the title of the paragraph as the readers might think that this is the only open question
- Deleted: ¶
- Formatted: Font: Not Italic
- Formatted: Font: Not Italic
- Deleted: engulfed
- Commented [SG72]: I don't understand, is this a real quote from darwin?
- Deleted: is
- Deleted: the origin of life
- Deleted: delicate
- Deleted: ¶
- It is fascinating that apparently all complex organisms on Earth probably go back to one crucial event, probably the merge of an alpha-proteobacterium and an Asgard archaeon, that occurred after approximately two billion years of independent evolution of two separate lineages, the bacteria and the archaea. ¶
- H
- Deleted: be repeated
- Deleted: ¶
- Deleted:
- Deleted: they are
- Deleted: ing
- Deleted: in
- Deleted: like
- Deleted: have
- Deleted: somewhat
- Deleted: ed
- Deleted: , more graspable
- Deleted: second
- Deleted: endo
- Deleted: with a bacterium
- Deleted: own being
- Deleted: a

889 Figure 2: Known morphological features of Asgard archaea. a-b. Scanning electron
micrographs of *Promethearchaeum syntrophicum*, the first cultured Asgard archaeon
(from Imachi et al., 2020) with potentially dividing cells in b, c. Transmission electron
micrograph of an ultrathin section of *P. syntrophicum* (Imachi et al., 2020), d-e. Scanning
electron micrographs of *Ca. Lokiarchaeum ossiferum*, the second cultured Asgard
archaeon (personal collection), f. Cryoelectron tomogram of *Ca. L. ossiferum* highlighting
the unusual cell surface structures of this organism (adapted from (Rodrigues-Oliveira et
al., 2023), g. CARD-FISH of *Lokiarchaeal* cells from sediments collected in Aarhus Bay,
Denmark. Red – probe Lok1183, Green – Probe Lok1378, Blue – DAPI (from Avci et al.,
2022), h. CARD-FISH of Heimdallarchaeal cells from sediments collected in Aarhus Bay,
Denmark. Red – probe Heim529, Green – Probe Heim329, Blue – DAPI (from Avci et al.,
2022), i. CARD-FISH of Hodarchaeales cells from sediments collected in Aarhus Bay,
Denmark. Red – probe Hod193, Green – Probe Hod286, Blue – DAPI (from Avci et al.,
2025). Scale bars: 1 μm (a, b, d, e, g, h, i), 500 nm (c) and 100 nm (f).

Deleted: divi

Formatted: Font: Italic

Deleted:

Formatted: Font: Italic

Deleted: l

Figures 3: Current hypothesis about the origin of eukaryotes, highlighting the central role
of Asgard archaea in this process and the early split of Bacteria and Archaea that both
contributed to the eukaryotic cell. Thin lines represent putative horizontal gene transfers
between domains.

Deleted: have

Deleted: simulate

Acknowledgements:

We would like to thank all colleagues who have contributed to the archaeal field over the
911 years, for their discussions and interactions. We apologize for not having cited everyone.

Deleted: T

Deleted: you to

Deleted: influenced

We are also indebted to Nathalia Jandl for help with references.

Commented [SG73]: how did she help?

Deleted: indebted

This work was supported by the Austrian Science Fund (FWF) with W1257 (Wittgenstein)
and EFP 25 (Emerging Field Project) to C.S.

References

Adl, S.M., Simpson, A.G., Lane, C.E., Lukes, J., Bass, D., Bowser, S.S., Brown, M.W., Burki,
F., Dunthorn, M., Hampl, V., Heiss, A., Hoppenrath, M., Lara, E., Le Gall, L., Lynn,
D.H., McManus, H., Mitchell, E.A., Mozley-Stanridge, S.E., Parfrey, L.W.,
Pawlowski, J., Rueckert, S., Shadwick, L., Schoch, C.L., Smirnov, A. and Spiegel,
F.W. 2012. The revised classification of eukaryotes. *J Eukaryot Microbiol* 59(5),
429-493.

Akil, C. and Robinson, R.C. 2018. Genomes of Asgard archaea encode profilins that
regulate actin. *Nature* 562(7727), 439-443.

Akil, C., Tran, L.T., Orhant-Prioux, M., Baskaran, Y., Manser, E., Blanchoin, L. and
Robinson, R.C. 2020. Insights into the evolution of regulated actin dynamics via
characterization of primitive gelsolin/cofilin proteins from Asgard archaea. *Proc*
*Natl Acad Sci U S A* 117(33), 19904-19913.

Albers, S.V., Forterre, P., Prangishvili, D. and Schleper, C. 2013. The legacy of Carl Woese
and Wolfram Zillig: from phylogeny to landmark discoveries. *Nat Rev Microbiol*
11(10), 713-719.

Albers, S.V. and Meyer, B.H. 2011. The archaeal cell envelope. *Nat Rev Microbiol* 9(6),
414-426.

Allers, T. and Mevarech, M. 2005. Archaeal genetics - the third way. *Nat Rev Genet* 6(1),
58-73.

Appler, K.E., Lingford, J.P., Gong, X., Panagiotou, K., Leao, P., Langwig, M.V., Greening,
C., Ettema, T.J., De Anda, V. and Baker, B.J. 2024. Oxygen metabolism in
descendants of the archaeal-eukaryotic ancestor. *bioRxiv*, 1-30.

Avci, B., Brandt, J., Nachmias, D., Elia, N., Albertsen, M., Ettema, T.J.G., Schramm, A. and
Kjeldsen, K.U. 2022. Spatial separation of ribosomes and DNA in Asgard archaeal
cells. *ISME J* 16(2), 606-610.

Avci, B., Panagiotou, K., Albertsen, M., Ettema, T.J.G., Schramm, A. and Kjeldsen, K.U.
2025. Peculiar morphology of Asgard archaeal cells close to the prokaryote-
eukaryote boundary. *mBio* 16(5), e0032725.

Aylett, C.H.S. and Duggin, I.G. (2017) Prokaryotic Cytoskeletons. *Subcellular*
*Biochemistry*. Löwe, J. and Amos, L.A. (eds), pp. 393-417, Springer, Cham.

Baker, B.J., De Anda, V., Seitz, K.W., Dombrowski, N., Santoro, A.E. and Lloyd, K.G. 2020.
Diversity, ecology and evolution of Archaea. *Nat Microbiol* 5(7), 887-900.

Ban, N., Beckmann, R., Cate, J.H., Dinman, J.D., Dragon, F., Ellis, S.R., Lafontaine, D.L.,
Lindahl, L., Liljas, A., Lipton, J.M., McAlear, M.A., Moore, P.B., Noller, H.F., Ortega,
961 J., Panse, V.G., Ramakrishnan, V., Spahn, C.M., Steitz, T.A., Tchorzewski, M.,
Tollervey, D., Warren, A.J., Williamson, J.R., Wilson, D., Yonath, A. and Yusupov,
963 M. 2014. A new system for naming ribosomal proteins. *Curr Opin Struct Biol* 24,
165-169.

Baquero, D.P., Borrel, G., Gazi, A., Martin-Gallausiaux, C., Cvirkaite-Krupovic, V.,
Commere, P.H., Pende, N., Tachon, S., Sartori-Rupp, A., Douche, T., Matondo, M.,
Gribaldo, S. and Krupovic, M. 2025. Biogenesis of DNA-carrying extracellular
vesicles by the dominant human gut methanogenic archaeon. *Nat Commun* 16(1),
5093.

Barry, E.R. and Bell, S.D. 2006. DNA replication in the archaea. *Microbiol Mol Biol Rev*
70(4), 876-887.

Baum, D.A. and Baum, B. 2014. An inside-out origin for the eukaryotic cell. *BMC Biol* 12,
76.

Bell, S.D. and Jackson, S.P. 1998. Transcription and translation in Archaea: a mosaic of
eukaryal and bacterial features. *Trends Microbiol* 6(6), 222-228.

Benelli, D., La Teana, A. and Londei, P. (2016) Evolution of the Protein Synthesis
Machinery and Its Regulation. Hernández, G. and Jagus, R. (eds), pp. 61-79,
Springer, Cham.

Benelli, D., La Teana, A. and Londei, P. (2017) RNA Metabolism and Gene Expression in
Archaea. *Nucleic Acids and Molecular Biology*. Clouet-d'Orval, B. (ed), Springer,
Cham.

Benelli, D. and Londei, P. 2011. Translation initiation in Archaea: conserved and domain-
specific features. *Biochem Soc Trans* 39(1), 89-93.

Bernander, R. and Ettema, T.J. 2010. FtsZ-less cell division in archaea and bacteria. *Curr*
*Opin Microbiol* 13(6), 747-752.

Blanch Jover, A. and Dekker, C. 2023. The archaeal Cdv cell division system. *Trends*
*Microbiol* 31(6), 601-615.

Blombach, F., Matelska, D., Fouqueau, T., Cackett, G. and Werner, F. 2019. Key
Concepts and Challenges in Archaeal Transcription. *J Mol Biol* 431(20), 4184-
4201.

Blombach, F., Smollett, K.L., Grohmann, D. and Werner, F. 2016. Molecular Mechanisms
of Transcription Initiation-Structure, Function, and Evolution of TFE/TFIIE-Like
Factors and Open Complex Formation. *J Mol Biol* 428(12), 2592-2606.

Bowman, J.C., Petrov, A.S., Frenkel-Pinter, M., Penev, P.I. and Williams, L.D. 2020. Root
of the Tree: The Significance, Evolution, and Origins of the Ribosome. *Chem Rev*
120(11), 4848-4878.

Bult, C.J., White, O., Olsen, G.J., Zhou, L., Fleischmann, R.D., Sutton, G.G., Blake, J.A.,
FitzGerald, L.M., Clayton, R.A., Gocayne, J.D., Kerlavage, A.R., Dougherty, B.A.,
Tomb, J.F., Adams, M.D., Reich, C.I., Overbeek, R., Kirkness, E.F., Weinstock,
1000 K.G., Merrick, J.M., Glodek, A., Scott, J.L., Geoghagen, N.S. and Venter, J.C. 1996.
Complete genome sequence of the methanogenic archaeon, *Methanococcus*
*jannaschii*. *Science* 273(5278), 1058-1073.

Bulzu, P.A., Andrei, A.S., Salcher, M.M., Mehrshad, M., Inoue, K., Kandori, H., Beja, O.,
Ghai, R. and Banciu, H.L. 2019. Casting light on Asgardarchaeota metabolism in
a sunlit microoxic niche. *Nat Microbiol* 4(7), 1129-1137.

Burki, F., Roger, A.J., Brown, M.W. and Simpson, A.G.B. 2020. The New Tree of
Eukaryotes. *Trends Ecol Evol* 35(1), 43-55.

Cai, M., Liu, Y., Yin, X., Zhou, Z., Friedrich, M.W., Richter-Heitmann, T., Nimzyk, R.,
Kulkarni, A., Wang, X., Li, W., Pan, J., Yang, Y., Gu, J.D. and Li, M. 2020. Diverse
Asgard archaea including the novel phylum Gerdarchaeota participate in organic
matter degradation. *Sci China Life Sci* 63(6), 886-897.

Cai, M., Richter-Heitmann, T., Yin, X., Huang, W.C., Yang, Y., Zhang, C., Duan, C., Pan, J.,
Liu, Y., Liu, Y., Friedrich, M.W. and Li, M. 2021. Ecological features and global
distribution of Asgard archaea. *Sci Total Environ* 758, 143581.

Carvalho, D.S., Andrade, R.F., Pinho, S.T., Goes-Neto, A., Lobao, T.C., Bomfim, G.C. and
El-Hani, C.N. 2015. What are the Evolutionary Origins of Mitochondria? A
Complex Network Approach. *PLoS One* 10(9), e0134988.

Costa, A., Hood, I.V. and Berger, J.M. 2013. Mechanisms for initiating cellular DNA
replication. *Annu Rev Biochem* 82, 25-54.

Coureux, P.D., Lazennec-Schurdevin, C., Bourcier, S., Mechulam, Y. and Schmitt, E.
2020. Cryo-EM study of an archaeal 30S initiation complex gives insights into
evolution of translation initiation. *Commun Biol* 3(1), 58.

Cox, C.J., Foster, P.G., Hirt, R.P., Harris, S.R. and Embley, T.M. 2008. The archaeobacterial
origin of eukaryotes. *Proc Natl Acad Sci U S A* 105(51), 20356-20361.

Cubonova, L., Richardson, T., Burkhart, B.W., Kelman, Z., Connolly, B.A., Reeve, J.N. and
Santangelo, T.J. 2013. Archaeal DNA polymerase D but not DNA polymerase B is

required for genome replication in *Thermococcus kodakarensis*. *J Bacteriol*
195(10), 2322-2328.

1029 Da Cunha, V., Gaia, M. and Forterre, P. 2022. The expanding Asgard archaea and their
elusive relationships with Eukarya. *mLife* 1(1), 3-12.

Darland, G., Brock, T.D., Samsonoff, W. and Conti, S.F. 1970. A thermophilic, acidophilic
mycoplasma isolated from a coal refuse pile. *Science* 170(3965), 1416-1418.

Darwin, C. 1871 Letter to J.D. Hooker. Hooker, J.D. (ed), University of Cambridge, Darwin
Correspondence Project.

DeLong, E.F. 1992. Archaea in coastal marine environments. *Proc Natl Acad Sci U S A*
89(12), 5685-5689.

Dennis, P.P. 1997. Ancient ciphers: translation in Archaea. *Cell* 89(7), 1007-1010.

Donoghue, P.C.J., Kay, C., Spang, A., Szollosi, G., Nenarokova, A., Moody, E.R.R., Pisani,
D. and Williams, T.A. 2023. Defining eukaryotes to dissect eukaryogenesis. *Curr*
*Biol* 33(17), R919-R929.

Duggin, I.G., Aylett, C.H., Walsh, J.C., Michie, K.A., Wang, Q., Turnbull, L., Dawson, E.M.,
Harry, E.J., Whitchurch, C.B., Amos, L.A. and Lowe, J. 2015. CetZ tubulin-like
proteins control archaeal cell shape. *Nature* 519(7543), 362-365.

Dyall-Smith, M. (2009) *The Halo handbook: Protocols for haloarchaeal genetics*, Ver 7.2.

Embley, T.M. and Martin, W. 2006. Eukaryotic evolution, changes and challenges. *Nature*
440(7084), 623-630.

Eme, L., Tamarit, D., Caceres, E.F., Stairs, C.W., De Anda, V., Schon, M.E., Seitz, K.W.,
Dombrowski, N., Lewis, W.H., Homa, F., Saw, J.H., Lombard, J., Nunoura, T., Li,
1049 W.J., Hua, Z.S., Chen, L.X., Banfield, J.F., John, E.S., Reysenbach, A.L., Stott, M.B.,
Schramm, A., Kjeldsen, K.U., Teske, A.P., Baker, B.J. and Ettema, T.J.G. 2023.
Inference and reconstruction of the heimdallarchaeal ancestry of eukaryotes.
*Nature* 618(7967), 992-999.

Feng, Y., Ding, J., Lin, Y., Cui, D., Li, K., Zheng, D., Cai, Z., Bell, S.D. and Wu, F. 2025.
Serial innovations by Asgard archaea shaped the DNA replication machinery of the
early eukaryotic ancestor. *Nat Ecol Evol*.

Forterre, P., Elie, C. and Kohiyama, M. 1984. Aphidicolin inhibits growth and DNA
synthesis in halophilic archaeobacteria. *J Bacteriol* 159(2), 800-802.

Fuhrman, J.A., McCallum, K. and Davis, A.A. 1992. Novel major archaeobacterial group
from marine plankton. *Nature* 356(6365), 148-149.

Gehring, A.M., Walker, J.E. and Santangelo, T.J. 2016. Transcription Regulation in
Archaea. *J Bacteriol* 198(14), 1906-1917.

Gerbi, S. (1996) *Ribosomal RNA: structure, evolution, processing, and function in protein*
*biosynthesis*. Zimmermann, R.A. and Dahlberg, A.E. (eds), pp. 71-87, Telford-CRC
Press, Boca Raton.

Grau-Bove, X., Navarrete, C., Chiva, C., Pribasni, T., Anto, M., Torruella, G., Galindo, L.J.,
Lang, B.F., Moreira, D., Lopez-Garcia, P., Ruiz-Trillo, I., Schleper, C., Sabido, E.
and Sebe-Pedros, A. 2022. A phylogenetic and proteomic reconstruction of
eukaryotic chromatin evolution. *Nat Ecol Evol* 6(7), 1007-1023.

Gribaldo, S., Poole, A.M., Daubin, V., Forterre, P. and Brochier-Armanet, C. 2010. The
origin of eukaryotes and their relationship with the Archaea: are we at a
phylogenomic impasse? *Nat Rev Microbiol* 8(10), 743-752.

Guy, L. and Ettema, T.J. 2011. The archaeal 'TACK' superphylum and the origin of
eukaryotes. *Trends Microbiol* 19(12), 580-587.

Point by point replies to Referee Reports

Referee #1:

This review represents an accessible, enjoyable and informative overview of progress in the archaeal field in general and Asgard in particular and is written by leading experts in the field. I think it will serve well as an introduction to this fascinating group of organisms. I particularly enjoyed the historical context in which the work was framed.

I have only a few specific comments.

Lines 155 - 157. The authors should explain that "coalescin" is a novel, archaeal specific SMC protein. Further, ClsN is enriched in transcriptionally less-active regions of the chromosome, not transcriptionally active regions as the authors state.

This has both been corrected.

Lines 197 - 199. There is no evidence that archaeal Orc1/Cdc6 form ring shaped hexamers. Rather the work of Samson et al (Mol. Cell, 2016) revealed, that in *Sulfolobus*, the Orc1-1 protein binds as a monomer to elements in the origin oriC1 and this monomeric Orc1-1, when bound to ATP, functions in both DNA binding and helicase recruitment roles.

This has been corrected.

Line 281-291

CdvA, B and C are not in an operon. CdvA is a separate transcription unit from the bicitronic cdvBC genes (see Samson, Mol Cell, 2011). The authors should re-phrase this section from "Encoded in an operon" to "encoded in a gene cluster". In addition, the authors should mention that CdvC is a homolog of eukaryotic Vps4 (which they discuss later in the context of Asgard ESCRT machineries). It's a shame that the CdvC nomenclature has been propagated in the literature, it just adds confusion to an already complex literature - indeed, the archaeal Vps4 was first named as Vps4 by Obita and colleagues in 2007 (Obita et al., Nature, 449, 735-739,) a year before Bernander and colleagues chose to rename it as CdvC.

It has been corrected

A minor point but the image in Figure 3 is awfully reminiscent of Jim Lake's Ring of Life hypothesis (proposed long before the discovery of Archaea". It would be nice to cite Lake's seminal proposal (Rivera and Lake Nature 2004 Sep 9;431(7005):152-5. doi: 10.1038/nature02848.)

This has been included in our paragraph on evolutionary models

I, Steve Bell, am happy to sign this report.

Referee #2:

Albeit surely not exhaustive (impossible task), this is an enjoyable and personal account that will surely be of interest to the readership of the journal. I had quite a few comments and suggestions that I made directly on the text. The authors can retain what they feel relevant and useful.

We are very grateful to Simonetta for going so carefully through the text!

The comments were extremely helpful and we implemented almost all of them.

There was also great input on the literature.

We followed the advice to add a tree for the nomenclature and phylogenetic relationships (now figure 1) and refer properly to all figures.

Referee #3

This review article by Schleper and Rodrigues-Oliveira represents a timely and (mostly) comprehensive overview of the history and current state-of-the-art of research into the archaea, and Asgard archaea more particularly. This covers a lot of ground, and does so in an even-handed way. The article will be invaluable for readers intrigued by the archaea. The figures are particularly attractive and I can see them being adapted for many lectures on archaea.

Inevitably, there are omissions. Several minor, some major. I have tried to outline these below. Overall, this is a very welcome and authoritative addition to the literature on the archaea / Asgard.

Thank you for this nice comment!

Major points

1. Many different archaeal clades, phyla and species are mentioned, but there's no figure to put this in perspective. Newcomers to the field could be very confused. What is a crenarchaeote? Euryarchaeote? TACK superphylum? How do they relate to Asgard or to each other? I feel another figure is essential.

Reviewer 2 made the same suggestion and we agree. So we have added a new figure (Now Figure 1) with all organisms that we cite in the text (and a few more for reference), placing them in the overall phylogeny/taxonomy. Since there are currently three parallel nomenclatures for archaea and readers want to be able to refer to older literature, we have decided to show only the 'order' levels as they mostly match in all taxonomies, and then add the superphyla by referring to the old namings and adding the new naming of the GTDB database that is used by most people. We state this clearly in the text and figure legend.

2. Another major omission is a discussion of the archaeal cell cycle - these needs at least a paragraph to do it justice - there are many interesting parallels and differences to eukaryotes in the different archaeal lineages.

Thank you for this suggestion, we have added a new paragraph on the cell cycle.

Specific points:

1. In several places, reviews are cited instead of the primary literature. The latter would be preferable, but if there's a lack of space then at least make it clear that the reference supplied is a review.

We have implemented this suggestion where, by referring to reviews as such when we cite those.

2. There is no discussion of Homologous Recombination / DSBR, which should be rectified if this is a comprehensive review.

We have added a few sentences also on DSBR now, however, covering the whole range of repair systems would mean to cover the diversity inside the archaea also, as they are not uniform. We had to keep it short for the balance of the review.

3. A general readership might be interested in the early research on archaeal CRISPR systems and the fact that archaea lack class 2 crspr (Cas9, Cas12) but are rich in type I and III.

This is an interesting topic, however, as we also don't cover the viruses, we can not cover the defense systems either. Also, we think it is a bit off topic, since CRISPR systems are mostly inherited by horizontal gene transfer among prokaryotes and this review focusses on the deep roots of eukaryotic features in archaea and including Asgard.

4. Methanogens are mentioned in several places, but readers may wish to know that all methanogens are archaeal, given the environmental implications.

A valid comment! A similar point was raised by reviewer 2. We have now dedicated a larger paragraph in the intro to the diversity of archaea, emphasizing also specifically the methanogens and their unique ecological role.

Minor points:

1. In general, the English could be improved a little. A few specific suggestions are given below.

The review has now been read by a native speaker.

2. Line 18. "It also.." - presumably this refers back to "The discovery" but the two clauses are separated by an entire paragraph, so maybe just re-state discovery in the final sentence.

has been fixed.

3. Line 21-23. Consider referencing one or more of Woese's papers here.

yes, we agree. has been done.

4. Line 33. The phrase "evolutionary achievements" might be considered a little unscientific?

has been exchanged

5. Line 35 "archaeal biology" fixed

6. Line 107 "of of" - fixed

7. Line 122-3. Consider adding the original references (Bell et al.) here. we have added a reference

8. Line 125 "an ATP hydrolysis" - step/reaction? fixed

9. Line 127 "these finding" -s fixed

10. Line 130 Blombach 2019 is a review - if you want to use this perhaps make this clear. – we have changed as suggested

11. Line 133 "recently discovered universally in archaea conserved" - need rephrased. fixed

12. Line 131, 135 - settle on one spelling of -og / -ogue - fixed, we settled on -og

13. Line 177 what does "and later also others" refer to? People or species?

has been clarified

14. Line 186-7 italicize species names. fixed

15. Line 187 sp "Archaeoglobus" fixed

16. Line 188. The phrase "which initiated biochemical studies" is confusing. Perhaps best deleted. - has been deleted

17. Line 213 - reference missing - fixed

18. Line 226 - references needed here, or at least a review. references were added

19. Line 256 typo Lokiarchaeia ? - taxonomic names have been carefully redone precipitating on one type of taxonomy for Asgard

20. Line 272 Define "TACK" done with figure

21. Line 512 - best not to mention unpublished data. we are citing now prepublished bioRxiv papers

22. Line 515 "asgard" fixed

23. Line 514-524 - needs some citations, even if they are repeated. fixed

Referee #4:

This is a timely review, that while accessible to newcomers to the field is nonetheless comprehensive and balanced. I have just a few comments.

Preamble

"It also raised novel hypotheses about the driving forces and mechanisms that gave rise to the emergence of the first eukaryotic cell" should be "The discovery of Asgard Archaea..."

fixed

"evolutionary achievements" - rephrase - done

"It is mainly addressed to newcomers" - should be "The review is meant primarily for newcomers to the field of archaeal molecular and cellular biology who..."

has been rephrased

"inventions" - should be "innovations" fixed

"ether bondage" -> "ether bond" fixed

"ester-linked fatty acids to a glycerol-3-phosphate" - rephrase - done

"racked the brains of researchers" -> "puzzled researchers" - done

"It will very interesting to see what kind of lipids the Asgard archaea have..." - actually Imachi et al., showed rather convincing evidence that 'MK-D1 probably contains C20-phytane and C40-biphytanes with 0-2 rings. The MK-D1 genome encoded most of the genes necessary to synthesize ether-type lipids-although geranylgeranylglyceryl phosphate synthase was missing-and lacked genes for ester-type lipid synthesis'

Yes there was an initial lipid study, but the lipid specialists in the field all agree that we need deeper biochemical analyses on this, because it was a mixed culture. We are on it together with Imachi also.

"ar7 for example from archaea" => "are archaea" or "are members of the archaea" - fixed

"i.e. thermophilic and thermoacidophilic" -> "i.e. hyperthermophilic and thermoacidophilic" - fixed

"N-terminal tales" -> |N-terminal tails" - fixed.

|between one copy to >20 ref." => fix -done

"Lokiarchaeia" -> check that it has not since also been renamed - yes we have now unified the namings

It is important to note that most current models of eukaryogenesis propose serial endosymbiosis, perhaps starting with deltaproteobacteria with alphaproteobacteria arriving much later, see recent work from the university of Bristol. These scenarios reject the "big bang" fusion that was popular before the cultivation of Asgard archaea. This aspect of the paper should be edited accordingly. - we have added the serial endosymbiosis this in the model section.

Dear Prof. Schleper,

I am pleased to inform you that your manuscript has been accepted for publication in the EMBO Journal.

If you have any questions, please do not hesitate to contact me or the Editorial Office. Thank you for your contribution to The EMBO Journal.

Yours sincerely,

Yehu Moran
Academic Editor
The EMBO Journal

Please note that it is The EMBO Journal policy for the transcript of the editorial process (containing referee reports and your response letters) to be published as an online supplement to each paper. If you should prefer removal of any referee-only figures included in the point-by-point response(s), e.g. because they may still be used for future publication or because they have been reproduced from published work by others, please do let us know immediately via response email.

More information is available here: <https://link.springer.com/partners/embo-press/editorial-policies#Peer%20review>